ecology/computational biology/mathematical modelling

gravity model, hierarchical model, infectious disease, invasive species, propagule pressure, zebra mussel

**Author for correspondence:**
S. M. Fischer
e-mail: samuel.fischer@ualberta.ca

# A hybrid gravity and route choice model to assess vector traffic in large-scale road networks

S. M. Fischer[1], M. Beck[3], L.-M. Herborg[4] and M. A. Lewis[1,2]

[1]Department of Mathematical and Statistical Sciences, and [2]Department of Biological Sciences, University of Alberta, Edmonton, Alberta, Canada
[3]Conservation Science Section, BC Ministry of Environment and Climate Change Strategy, Victoria, British Columbia, Canada
[4]Institute of Ocean Sciences, Fisheries and Oceans Canada, Sidney, British Columbia, Canada

 SMF, 0000-0001-8913-9575; MAL, 0000-0002-7155-7426

Human traffic along roads can be a major vector for infectious diseases and invasive species. Though most road traffic is local, a small number of long-distance trips can suffice to move an invasion or disease front forward. Therefore, understanding how many agents travel over long distances and which routes they choose is key to successful management of diseases and invasions. Stochastic gravity models have been used to estimate the distribution of trips between origins and destinations of agents. However, in large-scale systems, it is hard to collect the data required to fit these models, as the number of long-distance travellers is small, and origins and destinations can have multiple access points. Therefore, gravity models often provide only relative measures of the agent flow. Furthermore, gravity models yield no insights into which roads agents use. We resolve these issues by combining a stochastic gravity model with a stochastic route choice model. Our hybrid model can be fitted to survey data collected at roads that are used by many long-distance travellers. This decreases the sampling effort, allows us to obtain absolute predictions of both vector pressure and pathways, and permits rigorous model validation. After introducing our approach in general terms, we demonstrate its benefits by applying it to the potential invasion of zebra and quagga mussels (*Dreissena* spp.) to the Canadian province British Columbia. The model yields an $R^2$-value of 0.73 for variance-corrected agent counts at survey locations.

# 1. Introduction

Assessing road traffic and the transportation of goods through road networks is key to understanding the impacts of human movement in the context of epidemiology and invasion biology. For example, animal transport and trade are major vectors for animal and human diseases [1]. Similarly, many invasive species spread by means of human traffic along roads. Examples include plant seeds contained in dirt on cars [2], insects carried in firewood of campers [3], baitfish carried by anglers [4] and aquatic invasive species 'hitchhiking' on trailered watercraft [5].

To understand and control these processes, scientists and managers need estimates of the traffic flows in road networks. There are two perspectives on modelling traffic flows: the supply/demand perspective [6] and the route choice perspective [7]. While models for supply and demand (or travel incentive and destination choice) measure the motivation for travel or transport, route choice models determine the pathways along which the travel or transport occurs. Individually, supply/demand models and route choice models provide powerful tools for estimating traffic flows. However, as we will show below, there are situations where a hybrid approach is desirable.

The distribution of trips between origins and destinations is often modelled with gravity models [8], which have two main sources of data: on-site surveys of individual agents taken at source/destination locations, or mail-out surveys collecting details of planned or past trips from potential travellers. In general, on-site surveys yield precise estimates of absolute traffic flows but are more expensive, unless the data are readily available e.g. through booking records. In contrast, mail-out surveys may be more subject to sampling error but less expensive. While both survey types are used for parametrizing gravity models, field surveys are typically necessary if absolute measures of traffic flows are needed.

A potential alternative approach is to sample the traffic flow at given locations on roads. This contrasts with the on-site survey approach described above, where agents are sampled at source or destination locations. In many situations, surveys conducted at intermediate roads can provide much more data than origin/destination sampling. For example, consider a region with 100 possible sources and a region with 100 possible destination locations, with two main routes connecting them. The number of agents travelling along any of these main roads will, on average, be 50 times higher than the number leaving from or arriving at any individual location. Therefore, when there are many possible origin and destination locations but few major routes linking them, the number of agents sampled at intermediate roads will far exceed the number sampled leaving origins or arriving at destinations.

Because of the large amounts of data potentially available along roads, it would be advantageous to use such data to parametrize gravity models. However, to the best of our knowledge, this has not yet been done. As the traffic flow through roads depends on travellers' route preferences, a hybrid approach, which links gravity models to route choice models, would be required. This is the approach taken in this paper.

## 1.1. Gravity models and large-scale systems

The main idea of gravity models is to estimate the number of trips between an origin and a destination location based on agents' tendency to start a trip at the origin (repulsiveness), their tendency to travel to the destination (attractiveness), and the distance between origin and destination. Based on this basic idea, variations on gravity models have been derived to increase their predictive accuracy and mechanistic validity, such as constrained gravity models [9] and stochastic gravity models [10]. In 'classical' gravity models, traffic flows are assumed to be deterministic, and variations in observed traffic are viewed as measurement error. By contrast, stochastic gravity models suppose that the traffic flow itself is a stochastic process. That is, properties of donor and recipient determine the *mean* traffic flow, whereas the *actual* traffic flow varies over time, following some stochastic distribution.

Though stochastic gravity models were originally developed in the context of economics [10], they have also been successfully applied in invasion ecology and epidemiology to model the traffic of potential invasive species or disease vectors [4,11–17]. The systems modelled in these studies had small or medium spatial scale. However, long-distance trips can occur sufficiently often to pose a considerable risk of introducing invasive species or diseases to regions far away from the infested area. Hence, long-distance trips can be a major factor for shifting invasion or disease fronts [18]. Therefore, models for long-distance traffic are needed.

In large-scale systems, it is hard to collect the data required to fit a gravity model. Often, origins and destinations span over large areas, or regions of origin and destination may be considered instead of

individual locations. In both cases, the considered origins and destinations have many access points, which are expensive to monitor all at once. Conducting mail-out surveys is usually not an option, too, as the number of agents who could *potentially* start a long-distance trip is large while only few *will* actually do so and thus provide useful data. For example, almost any boat owner could potentially take their boat on a 20-hour journey, thereby transporting aquatic invasive species over long distances. Nonetheless, only few boaters ever make such a long journey, and identifying them for a survey is challenging. Due to these reasons, an alternative approach to mail-out surveys or origin/destination-based sampling is required to fit gravity models in large-scale systems.

The shortcomings of gravity models in large-scale systems concern not only the model fit but also how the models can be used to facilitate management of diseases or invasive species. A common management goal is to reduce the number of vectors leaving an infested area or entering a susceptible area. As the number of origins and destinations is large and they may have many access points in large-scale systems, it may be infeasible to apply control directly at the infested and susceptible locations. Instead, managers may want to control the traffic on intermediate roads that are shared by agents travelling from different origins to different destinations. To find the best roads for such control measures, a route choice model is necessary, which determines how the traffic between an origin and a destination is distributed over the road network.

## 1.2. Route choice models

Travellers are usually not able to consider all possible routes to their destination due to the vast number of options. Therefore, many route choice models assume that travellers make route choices in two steps: first, they apply some heuristic to determine a set of potentially good (admissible) routes, and second, they choose one of these routes based on the routes' characteristics [19].

A variety of approaches have been developed to model the two decision steps. Models for route admissibility may determine all routes that satisfy certain criteria or focus on routes that are optimal with respect to different goodness measures [20]. Alternatively, locally optimal routes may be considered [21,22], which assume that travellers act rationally on local scales while unknown factors may affect the routes on large scales. This method has been found to yield realistic routes while maintaining high computational efficiency [21,22].

To model the second stage of the decision process, the admissible routes are typically assigned probabilities for being chosen. The corresponding models may include economic aspects, such as the length of a route and the expected travel time, but also other factors, such as potential intermediate destinations and the scenery and sights along a route [7]. However, since multiple admissible routes between all combinations of origins and destinations must be considered, large-scale systems require a model balancing accuracy with computational efficiency.

## 1.3. Outline

Both gravity models and route choice models are widely used in their respective fields. In this paper, we present a hybrid model combining the two to assess traffic in large-scale systems. Since traffic varies over time, we use an additional model to account for time-driven variations in survey data. Furthermore, we introduce another model for the compliance of travellers, because not every traveller may participate in the survey and provide complete information. This hybrid approach allows us to fit a gravity model to data collected in roadside surveys. As a result, the hybrid method is applicable regardless of the system's spatial scale and yields not only estimates of the traffic outflow and inflow of origins and destinations but also estimates the traffic volume on roads.

We demonstrate our approach by applying it to the potential invasion of zebra and quagga mussels *Dreissena* spp. to the Canadian province British Columbia (BC). Dreissenid mussels are invasive in North America and cause severe economic and ecological damages [23,24]. A major spread mechanism of zebra and quagga mussels is boaters transporting mussel-infested watercraft and gear to uninvaded lakes [5]. Therefore, knowledge of destinations and travel routes for these boaters is key for mussel prevention and early detection.

This paper is structured as follows: in §2, we give an overview of the hybrid approach and the submodels for the the distribution of trips between origins and destinations, the route choice, temporal traffic patterns and the compliance of travellers. In §3, we describe how survey data collected at roads can be used to fit the submodels. In §4, we apply the hybrid model to the potential invasion of dreissenid mussels to BC and present the resulting estimates of vector pressure and pathways in BC. Finally, in §5, we discuss shortcomings, applicability and potential extensions of our approach.

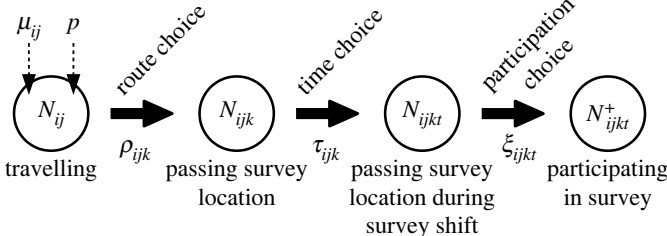

**Figure 1.** Hierarchical stochastic model for the number of agents passing a survey location during a survey shift. The total number $N_{ij}$ of agents travelling from $i$ to $j$ depends on the parameters $\mu_{ij}$ and $p$. With a probability $\rho_{ijk}$, the travelling agents will choose a route via the survey location $k$. With probability $\tau_{ijkt}$, the $N_{ijk}$ agents who choose such a route will also time their journey so that they pass the location in the time interval $t$ when the survey is conducted. These $N_{ijkt}$ agents choose with probability $\xi_{ijkt}$ to participate in the survey and to provide complete information. The resulting $N_{ijkt}^{+}$ agents are the ones included in the survey.

# 2. Model

Before introducing our hybrid traffic model, we need to clarify which travellers we want to consider. Not every person travelling from an infested region to a susceptible destination has the potential to carry a disease or invasive species. Similarly, not every potential carrier of propagules or pathogens will actually be infested and thus be a vector. In this paper, we assess the traffic of all *potential* vectors, regardless of whether they carry pathogens or propagules. Below, we call these potential vectors 'agents'.

We propose a hierarchical approach to model how many agents can be observed in a survey shift conducted at a road side. An agent will be observed in a roadside survey if, and only if, they (i) start a trip, (ii) choose a route via the survey location, (iii) time their journey so that they pass the survey location during the survey shift, and (iv) participate in the survey. Since these decisions are difficult to know precisely, we assume that the number of surveyed agents results from a hierarchical stochastic process (figure 1): (i) every time unit, a random number $N_{ij}$ of agents travel from origin $i$ to destination $j$; (ii) out of these agents, a random number $N_{ijk}$ choose a route via the survey location $k$; (iii) out of these agents, a random number $N_{ijkt}$ time their journey so that they pass the survey location during the time interval $t$ when the survey is conducted; (iv) out of these agents, a random number $N_{ijkt}^{+}$ agents decide to participate in the survey and provide complete information. This approach allows us to fit the model to data collected in roadside surveys.

The distributions of $N_{ij}$, $N_{ijk}$, $N_{ijkt}$ and $N_{ijkt}^{+}$ depend on submodels. Though some applications may require more specific submodels, we now propose a set of models applicable in many real-world systems. A detailed list of our assumptions can be found in electronic supplementary material, appendix A.

## 2.1. Gravity model

We model the daily numbers $N_{ij}$ of agents travelling from origin $i$ to destination $j$ with a stochastic gravity model. The mean value $\mu_{ij}$ of the random variable $N_{ij}$ is proportional to the repulsiveness $m_i$ of the origin $i$, the attractiveness $a_j$ of the destination $j$, and a negative power of the distance $d_{ij}$ between $i$ and $j$,

$$\mu_{ij} = c \frac{m_i a_j}{d_{ij}^{\alpha_d}}. \tag{2.1}$$

Usually, $m_i$ and $a_j$ are estimated as functions of covariates that correlate with the number of agents leaving donor region $i$ and the number of agents arriving at recipient $j$, respectively. The constant $c$ is a scaling factor.

The functions used to estimate $m_i$ and $a_j$ consist of 'building blocks' corresponding to one covariate $x_r$, $r \in \{1, \ldots, n\}$, each. Convenient functional forms for the building blocks are the power function $f_0(x_r) := x_r^{\alpha_1}$ and the saturating function $f_1(x_r) := (x_r/(x_r + \alpha_0))^{\alpha_1}$. The functional form $f_1$ is appropriate if the covariate has a particularly high impact after some threshold value or if differences in large covariate values are insignificant (e.g. [12]). Otherwise, $f_0$ is typically sufficient.

Many such building blocks can be connected to account for spatial heterogeneity. If two covariates are effective only in combination with each other, their respective building blocks should be multiplied together. For example, if *both* recreational opportunities and accommodations are necessary

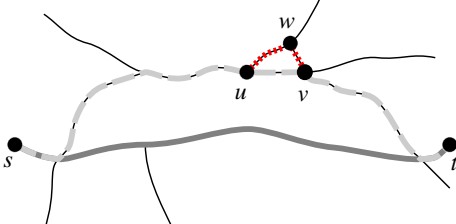

**Figure 2.** Admissible paths from origin $s$ to destination $t$. For a moderate choice of $\delta$, the shortest path (solid grey) and the path via $u$ and $v$ (dashed grey) are admissible. The path via point $w$ (dotted red from $u$ to $v$, dashed grey from $s$ to $u$ and from $v$ to $t$) is inadmissible, because it is not locally optimal: the short subsection $u \rightarrow w \rightarrow v$ (dotted red) is not a shortest path. The quantitative definition of 'short' is controlled via the parameter $\delta$. With $\delta = 0$, all shown paths would be admissible, whereas with $\delta = 1$, only the shortest path (solid dark grey) would be admissible.

to attract agents, attractiveness is given by the product of the corresponding building blocks. In turn, if covariates have an effect independent of each other, the respective building blocks should be added together. For example, if *either* a boat launch or mountain biking opportunities can attract agents, the corresponding building blocks should be added together. In that sense, multiplication models an 'and' relationship, whereas addition models an 'or' relationship. More complicated interactions can be modelled by adding together different combinations of multiplied building blocks.

Though the mean number $\mu_{ij}$ of travelling agents is given by a deterministic function, the number $N_{ij}$ of agents travelling in a time unit follows a stochastic distribution. Most stochastic gravity models build on the Poisson distribution, the negative binomial distribution or the zero-inflated negative binomial distribution [25]. The Poisson distribution is applicable if agents decide independently of each other in each time unit whether they start a trip. If agents' decisions are correlated, for example because weather conditions, holidays and other factors affect many agents at once, the density of the Poisson random variable can be chosen to vary with time. If the sources of correlations are not known precisely, a negative binomial distribution can be used to approximately account for the overdispersion resulting from such correlations [26]. Lastly, zero-inflated distributions suppose that there is a stochastic mechanism that stops all agents from travelling between an origin and a destination in some time units. In the remaining time units, $N_{ij}$ is assumed to follow a common stochastic distribution, such as the negative binomial distribution. We build our gravity model based on the negative binomial distribution, as this distribution is appropriate in many use cases and generalizes the Poisson distribution.

We parametrize the count distribution so that the ratio between mean and variance of the agent counts is constant for all origin–destination pairs. With this parametrization, the sum of two independent negative binomial random variables is still negative binomially distributed. This is particularly important when the model is built to assess traffic between regions of multiple individual origin or destination locations. In this scenario, the flow between the regions is the sum of the flows between the individual locations. Choosing a constant mean to variance ratio makes the model invariant to how the individual locations are pooled together. Refer to electronic supplementary material, appendix B, for further details.

## 2.2. Route choice model

We assume that agents choose their routes randomly and independently from one another. Though stochastic traffic events such as traffic jams could affect multiple agents and thus be a source of correlations, these events often affect route choices on small scales only and are thus unlikely to affect agent counts at specific locations significantly. Furthermore, agents of concern usually constitute only a fraction of the full traffic on a road. Therefore, traffic jams and other traffic-dependent factors affecting route attractiveness are mostly independent of the modelled agents' routing decisions, so that these decisions rarely affect each other directly.

Many route choice models assume that agents choose their routes from a small set of 'admissible' routes [7]. We define route admissibility following Fischer [22], who assumes that admissible paths do not contain local detours. The rationale behind this assumption is that major route decisions may be affected by factors unknown to us, while minor route decisions follow strict rational rules. Consequently, an admissible path $P$ can only contain a detour if the detour is longer than $\delta \cdot \text{length}(P)$. The constant $\delta$ defines which detours are deemed 'local'. We illustrate this concept of local optimality in figure 2.

The resulting set of admissible paths may still be very large. To limit the number of admissible paths further, we require that they are not more than a factor $\gamma$ longer than the shortest alternative. Constraining the length of *entire paths* contrasts with the local optimality constraint, which discards paths with *local detours*. In addition to limiting path lengths, we focus on 'single-via paths' only. These are shortest paths via one arbitrary intermediate destination, respectively. Without the latter constraint, the set of admissible paths could be intractably large and include unrealistic zigzag routes [22]. We compute the set of admissible paths with the algorithm by Fischer [22].

After computing the set of paths that agents may choose from, we need to assign the individual paths with probabilities. We assume that the probability that an agent chooses a route $P$ is inversely proportional to a power of its length $l_P$. That is, if $\mathcal{P}_{ij}$ is the set of admissible routes from origin $i$ to destination $j$ and $\lambda \geq 0$ a constant, the probability to choose route $P$ is given by

$$\mathbb{P}(\text{choose route } P \mid \text{travelling on admissible route}) = \frac{l_P^{-\lambda}}{\sum_{\tilde{P} \in \mathcal{P}_{ij}} l_{\tilde{P}}^{-\lambda}}. \tag{2.2}$$

Though we expect most agents to drive on admissible paths, some agents may choose routes deemed inadmissible. We account for that possibility by assuming that agents choose inadmissible routes with a small probability $\eta_c$. As these agents could choose any path through the road network, it is difficult to estimate the probability to observe such agents at a specific survey location. In the absence of a 'good' model and considering that only few agents choose inadmissible routes, we assume that any survey location could be on any inadmissible route with probability $\eta_o$, respectively. In summary, the probability that an agent travelling from $i$ to $j$ passes a survey location $k$ is

$$\rho_{ijk} = \underbrace{(1 - \eta_c)}_{\substack{\text{prob. to choose} \\ \text{an adm. route}}} \underbrace{\sum_{P \in P_{ij}: k \in P}}_{\substack{\text{sum over all} \\ \text{adm. routes via } k}} \underbrace{\frac{l_P^{-\lambda}}{\sum_{\tilde{P} \in P_{ij}} l_{\tilde{P}}^{-\lambda}}}_{\substack{\text{prob. to} \\ \text{choose route } P}} + \underbrace{\eta_c \eta_o}_{\substack{\text{prob. to be observed} \\ \text{on inadm. route}}} \tag{2.3}$$

## 2.3. Temporal pattern model

The numbers of agents observed in roadside surveys vary in temporal patterns. These traffic patterns must be accounted for to (i) avoid the introduction of a bias if not all traffic surveys have been conducted in similar time intervals, and to (ii) extrapolate part-time observations to the full time. Traffic may fluctuate in daily, weekly, and seasonal cycles and depend on the survey location, because agents will reach locations far away from their starting points later than locations close to their origins. In this study, we focus on daily patterns to keep the model simple. Furthermore, we assume that the temporal traffic pattern is independent of the survey location, because starting time, travel speed and overnight breaks vary among agents. The complex interplay of these factors makes it difficult to model traffic patterns mechanistically. Therefore, it is appropriate to use a simple phenomenological traffic pattern model.

Unimodal cyclic distributions constitute a good first approximation to daily traffic patterns, since traffic is denser during the day than during the night, in general. A commonly used unimodal cyclic distribution is the von Mises distribution [27]. This distribution resembles a normal distribution and takes a location parameter, determining the traffic peak time and a scale parameter, controlling how sharp the peak is. Other distributions can be used if traffic is expected to follow a more complex pattern, but we will proceed with the von Mises distribution due to its simplicity and intuitive shape.

## 2.4. Compliance model

The number of agents stopping to be surveyed may depend on their origin and destination, the time of day of the survey, and the set-up of the survey location. For example, more agents may stop if the survey location is clearly visible or if compliance can be enforced. If required, the compliance rate could be measured for each survey location individually. However, to keep the model simple, we assume that the probability that an agent chooses to participate in the survey and provides complete and correct data is constant across agents, survey time and survey locations.

# 3. Model fit

In the previous section, we described a hierarchical model for the number of agents observed in a roadside survey shift. In this section, we show how such survey data can be used to fit the model.

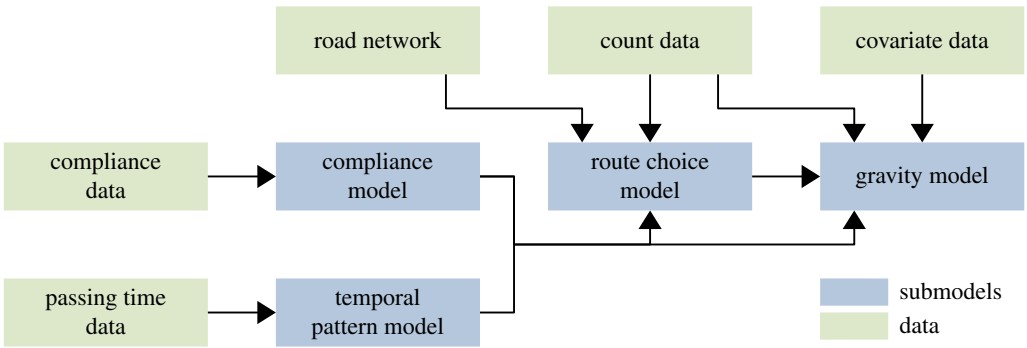

**Figure 3.** Overview of the model fitting procedure. The green rectangles depict data; the blue rectangles depict submodels. The arrows show which components are needed to fit the four submodels, respectively.

We fit the four submodels in the order inverse to the hierarchy. That is, we start with the compliance model and the temporal pattern model, proceed with the route choice model, and end with the gravity model (figure 3). Before we describe the fitting procedures in detail, we give an overview of the data required to fit the model.

## 3.1. Required data

We need five datasets to fit our hybrid model: (i) a count dataset, (ii) a compliance dataset, (iii) a survey time dataset, (iv) a covariate dataset, and (v) a graph representation of the road network with edges weighted by length or travel time. The count dataset contains the start and end time of each survey shift, the respective survey location and how many agents were surveyed driving from each origin to each destination. Most of these count values will be zero, especially if many origin–destination pairs are considered. The compliance dataset contains the total number of agents who passed the survey locations and the number of agents who participated in the survey and provided complete data. The survey time dataset encompasses the times of day when agents were surveyed and the start and end times of the respective survey shifts. The covariate dataset contains information related to the outbound and inbound traffic volume at origins and destinations. For example, this could be the population counts for the source locations or the number of close-by tourist attractions for the destination locations. Lastly, we require a graph representation of the considered road network. Roads translate to edges, weighted by the roads' respective lengths or the time required to drive along the roads. The set of vertices consists of all junctions of the road network as well as the origins and destinations of the agents. All survey locations, origins and destinations must correspond to specific vertices or edges in the graph. Collectively, the five datasets are shown by the green rectangles in figure 3.

## 3.2. Fitting the compliance model

The compliance model measures which proportion of agents is expected to stop at a survey location and to provide complete and consistent data. The model can be fitted in a single step by dividing the number of agents who provided good data by the total number of agents who passed the survey locations. However, in some applications, the origin of agents and other information impacting their potential of being a vector can be determined easily as soon as they stop for the survey. This could be done, for example, by using licence plate information. If this is possible, only the data provided by potentially infested agents needs to be checked for integrity, and the overall compliance rate $\xi$ may be obtained by estimating the participation rate $\xi_p$ and the complete/consistent data rate $\xi_c$ separately.

We compute the rate $\xi_p$ using count data of how many agents stopped at survey locations and how many agents passed these locations without stopping. The estimated participation rate $\xi_p$ is given by

$$\xi_p = \frac{\#\text{agents stopped}}{\#\text{agents stopped} + \#\text{agents bypassed}}.$$ (3.1)

Similarly, we compute the complete/consistent data rate $\xi_c$ as

$$\xi_c = \frac{\#\text{high-risk agents providing complete and consistent data}}{\#\text{high-risk agents stopped}}.$$ (3.2)

The overall compliance rate $\xi$ is the product of the two rates,

$$\xi = \xi_p \xi_c. \tag{3.3}$$

## 3.3. Fitting the temporal pattern model

The temporal pattern model accounts for the temporal variations in the traffic density. When we fit this model, we have to take into account that the survey shifts in which the data were collected do not cover all times of day equally well in general. For example, if most surveys were conducted in the morning, our dataset would contain a disproportionate number of agents observed in the morning, even if the true traffic peak were during the afternoon. To avoid the resulting bias, we fit our model with a maximum-likelihood approach based on the conditional likelihood, which takes into account when the surveys were conducted. We provide details in electronic supplementary material, appendix D.

## 3.4. Fitting the route choice model

The route choice model specifies the probabilities that agents take specific routes. As with the temporal pattern model, we fit the route choice model based on the conditional likelihood. Usually, it is infeasible to monitor all potential routes of agents at once, and surveyors have to focus on a small set of routes. To ensure that our choice of survey locations does not bias our results, we fit the route choice model by maximizing the likelihood conditional on which routes we monitored for how long.

There are several practical challenges associated with fitting the route choice model. These challenges are not only due to the computational complexity of the task but also due to identifiability problems, which could lead to non-informative results. In particular, the probability $\eta_c$ that boaters choose an inadmissible route and the probability $\eta_o$ that such boaters pass a survey location are not estimable, because we do not know how many agents travel along inadmissible routes that were not covered by any survey station. Therefore, we bound the traffic on inadmissible routes via an additional assumption constraining the probability $\eta_c$ that agents choose inadmissible routes to $\eta_c \leq 0.05$. In electronic supplementary material, appendix D, we provide more details.

## 3.5. Fitting the gravity model

The gravity model estimates how many agents are driving from each origin to each destination per time unit. We fit the model by maximizing the composite likelihood [28]. The difference with classical likelihood estimation is that we make an approximation via independence assumptions so as to facilitate straightforward computation.

When we fit the gravity model, we exploit that the number $N_{ijkt}^+$ of surveyed agents is negative binomially distributed [29]. This simplifies the model fit, as the likelihood function can be written down easily. Nonetheless, computing the likelihood is computationally costly, because each survey shift yields a count value for each origin–destination pair. In electronic supplementary material, appendix D, we present an algorithm to speed up the computations by orders of magnitude.

# 4. Application

In the previous sections, we outlined the hybrid gravity, route choice, temporal pattern and compliance model, and described how it can be fitted to data. Now we demonstrate our approach by applying it to the potential invasion of zebra and quagga mussels *Dreissena* spp. to the Canadian province BC.

## 4.1. Methods

We fit the hybrid model with survey data collected by the BC Invasive Mussel Defence Program. The survey data were obtained during 1571 survey shifts at 31 locations in BC over the course of the years 2015 and 2016. All shifts were conducted during day time. As small boats present a lower risk of being fouled by dreissenid mussels, we counted only medium to large motorized watercraft (e.g. cabin cruiser, wakeboard boats, speed boats, car toppers) as potential mussel vectors.

By provincial law, it was mandatory for boaters to stop at the survey locations. Nonetheless, not all boaters complied with this provision. The number of bypassing boaters was counted in 293 of the survey

shifts. As it was difficult to determine the type of bypassing towed boats precisely, we did not distinguish between boat types when estimating the participation rate. However, when we determined the proportion of boaters providing consistent and complete data, we excluded low-risk watercraft and local traffic.

We identified 5981 potentially boater accessible lakes in BC and considered them as potential destinations for the boaters. As origins we included the Canadian provinces and territories and the American states of the North American mainland. We treated a state or province as potential zebra and quagga mussel donor if either (i) there was a confirmed dreissenid mussel detection in a waterbody within the jurisdiction, or (ii) if the jurisdiction (iia) had a connected waterway with a dreissenid mussel infested lake in a neighbouring state or province and (iib) did not have an established dreissenid mussel monitoring programme at the time the data were collected. All remaining source jurisdictions were used to fit the model but ignored when we assessed potential propagule transport.

To increase the spatial resolution for the boater origins close to BC, we split the province Alberta into three parts (north, middle and south) and the state Washington into an eastern and a western part. Similarly, we split certain large lakes. Some lakes in BC span hundreds of kilometres. This can make it difficult to determine the best access routes if the lakes have far-apart access points. Therefore, we checked the access routes to all lakes with a perimeter larger than 100 km and split the lakes that were accessible via multiple substantially different routes.

The model and the fitting methods were implemented in Python 3.7 in combination with Scipy 1.0 [30] and Cython 0.26 [31]. Our code is publicly available at vemomoto.github.io.

### 4.1.1. Structure of the gravity model

To estimate the repulsiveness of donors, we assumed that the nation and the boater count of a jurisdiction act together in yielding high counts of travelling boaters. As the number of boaters residing in the jurisdictions is unknown, we tested both population and angler number as proxies for the boater number. To estimate lake attractiveness, we considered the lake area, the lake perimeter, the presence of marinas, campgrounds and other facilities (including public toilets, tourist information, viewpoints, parks, attractions and picnic sites) in a 500 m range of the lakes, and the population living in 5 km ranges around the lakes. Thereby, we assumed that both a sufficient size and the presence of tourist facilities are necessary to attract many boaters, whereby the type of the facilities is of minor importance. We tested both lake area and lake perimeter as measures for the lake size. To measure distances and compute potential routes, we used a road network with edges weighed based on travel time. We provide further details of the data, including a list of the data sources, in electronic supplementary material, appendix C.

Connecting all building blocks, we arrived at the following model for the daily mean number of travelling agents:

$$\mu_{ij} = c \cdot \left( \frac{\text{pop}_i}{\text{pop}_i + \text{pop}_0} \right)^{\alpha_{\text{pop}}} \cdot \beta_{\text{CA}}^{\text{CA}_i} \cdot \left( \frac{A_j}{A_j + A_0} \right)^{\alpha_{\text{A}}}$$
$$\cdot \left( 1 + \beta_{\text{camp}} \text{camp}_j + \beta_{\text{fac}} \text{fac}_j + \beta_{\text{mar}} \text{mar}_j + \beta_{\text{lpop}} \left( \frac{\text{lpop}_j}{\text{lpop}_j + \text{lpop}_0} \right)^{\alpha_{\text{lpop}}} \right) \cdot d_{ij}^{-\alpha_{\text{d}}}. \tag{4.1}$$

Refer to table 1 for an explanation of the symbols.

### 4.1.2. Model selection and validation

We used a model selection criterion to determine which covariates our model should include to fit the data well without overfitting. Contrasting the criterion by Akaike (AIC) and the Bayesian information criterion (BIC), Ghosh & Samanta [32] point out that AIC is to be preferred if the goal is to provide precise predictions. Therefore, we selected our model based on AIC. See electronic supplementary material, appendix E for a more in-depth discussion of model selection.

Our model candidates incorporated a large number of covariates. Therefore, it was not feasible to check all possible combinations of covariates, parameters and functional forms of the building blocks. Thus, we ignored models with few covariates after noting that these models had much larger AIC values in general.

**Table 1.** Covariates, parameters and estimated parameter values along with 95% confidence intervals for the best-fitting gravity model. Refer to electronic supplementary material, appendix G, for a discussion of the large confidence intervals for $\beta_{lpop}$ and $lpop_0$. Parameters without confidence intervals ('–') were not part of the model with the best AIC value and fixed beforehand. Further covariates tested but not included in the model with the best AIC value were angler counts in jurisdictions and lake perimeters. These covariates were tested in place of the covariates $pop_i$ and $A_j$, respectively.

| covariate | explanation | parameter | estimate | profile CI | |
|---|---|---|---|---|---|
| — | scaling factor | $c$ | $3.73e_{-8}$ | $2.36e_{-8}$ | $5.83e_{-8}$ |
| — | mean/variance | $p$ | 0.23 | 0.21 | 0.25 |
| $pop_i$ | population of jurisdiction $i$ ($1e_6$) | $pop_0$ | 0.16 | 0.09 | 0.26 |
| | | $\alpha_{pop}$ | 1 | — | — |
| $CA_i$ | 1 if jurisdiction $i$ is Canadian, else 0 | $\beta_{CA}$ | 14.79 | 12.82 | 17.15 |
| $camp_j$ | 1 if major campgrounds are present at lake $j$, else 0 | $\beta_{camp}$ | 6.55 | 4.66 | 9.3 |
| $fac_j$ | 1 if other facilities (toilets, viewpoints, tourist infos, parks, attractions, picnic sites) are present at lake $j$, else 0 | $\beta_{fac}$ | 4.51 | 3.04 | 6.63 |
| $mar_j$ | 1 if marinas are present at lake $j$, else 0 | $\beta_{mar}$ | 26.4 | 19.41 | 36.59 |
| $lpop_j$ | population living closer than 5 km to the lake $j$ ($1e_3$) | $\beta_{lpop}$ | 1011 | 396 | $>1e_{10}$ |
| | | $lpop_0$ | 888 | 318 | $>1e_{10}$ |
| | | $\alpha_{lpop}$ | 1 | — | — |
| $A_j$ | area of lake $j$ ($km^2$) | $A_0$ | 1236 | 1044 | 1464 |
| | | $\alpha_A$ | 1 | — | — |
| $d_{ij}$ | shortest traveltime between jurisdiction $i$ and lake $j$ ($1e_4$ min) | $\alpha_d$ | 3.45 | 3.35 | 3.54 |

To get a sense of the credibility of our parameter estimates and check for estimability issues, we determined confidence intervals for the model parameters based on the profile likelihood [33,34] (see also our notes on composite likelihood-based confidence intervals in electronic supplementary material, appendix E). Since correlations between predictors cause wide parameter confidence intervals [35], confidence intervals also yield insights into whether multicollinearity between covariates is an issue. Besides studying the confidence intervals of model parameters, we tested our base hypotheses on boater counts and the temporal traffic pattern. Details can be found in electronic supplementary material, appendix F.

To assess the predictive capability of our model, we compared observed count data to the corresponding model predictions. Thereby, we focused on the traffic flow at inspection stations, as this quantity is of high management interest and we had enough data to conduct a thorough analysis. We considered data collected between 11.00 and 16.00, so that the count data were identically distributed for each survey location, respectively. As a result, the mean traffic estimates were approximately normally distributed according to the central limit theorem. To account for heteroscedasticity, we normalized observations and predictions by their predicted standard deviations. This allowed us to determine the $R^2$-value as a well-known measure for predictive power.

We applied this analysis not only to the dataset used to fit the model but also to a distinct validation dataset. To generate the validation dataset, we randomly selected 30% of all survey shifts. The remaining data were used to fit the model. More details and further validation results are provided in electronic supplementary material, appendix F.

The $R^2$-value described above gives insight into the model's ability to predict the mean boater counts at roads. To assess the overall model fit, we also computed the pseudo $R^2$ suggested by Nagelkerke [36]. This metric compares the likelihood of the fitted model to the likelihood of a null model and reduces to the classical $R^2$ in linear regression analysis [36]. As null model, we used our hybrid model with a uniform temporal traffic distribution and a route choice model that does not take into account any route information

**Table 2.** Parameters and estimates along with 95% confidence intervals for the route choice model. As $\eta_c$ and $\eta_o$ are not estimable, we bounded $\eta_c \leq 0.05$ to obtain the final parameter estimates (see electronic supplementary material, appendix D). Since the likelihood function is not continuous in the parameters $\gamma$ and $\delta$ and computing admissible routes is computationally expensive, we did not construct confidence intervals for these parameters.

| parameter | explanation | estimate | profile CI | |
|---|---|---|---|---|
| $\gamma$ | maximal stretch of admissible paths | 1.4 | — | — |
| $\delta$ | required local optimality of admissible paths | 0.2 | — | — |
| $\eta_c$ | probability to travel on an inadmissible path | 0.049 | 0.013 | 0.05 |
| $\eta_o$ | probability to pass a given survey location if travelling on an inadmissible path | 0.062 | 0.044 | 0.47 |
| $\lambda$ | travel time exponent | 7.4 | 6.53 | 8.29 |

(i.e. $\eta_c = 1$). The corresponding gravity model included a constant mean $\mu_{ij}$ and a constant mean to variance ratio for all origin–destination pairs. Since Nagelkerke's pseudo $R^2$ is based on the likelihood, it does not require that the considered data are normally distributed with equal variance.

## 4.2. Results

In this section, we provide information on the fitted submodels and show results on the compliance rate, the temporal traffic distribution, the sources of high-risk boaters, the boater inflow to susceptible lakes and the boater traffic through the road network. Furthermore, we present model validation results.

### 4.2.1. Resulting models

*Gravity model*. The gravity model with minimal AIC value included eight covariates and 11 parameters. The parameter values can be found in table 1 along with their confidence intervals. Since the gravity model is a phenomenological model, the parameter values have limited meaning. Nonetheless, we can make some comparative statements regarding the roles of the different covariates in our model.

The repulsiveness $m_i$ of source jurisdictions was estimated based on their population count and nation. Canadian provinces were weighed about 15 times higher than American states. The numbers of anglers in the jurisdictions were not included.

The submodel for the lake attractiveness $a_j$ included the covariates lake area, presence of campgrounds, marinas and other facilities, and the population living close to the lakes. The sole presence of 'other facilities' (public toilets, viewpoints, etc.; see table 1) increased the attractiveness of a lake by factor 5.51. The presence of campgrounds weighed 45% more than the presence of these facilities. The presence of a marina, in turn, weighed more than four times as much as the presence of a campground. An equally important factor for lake attractiveness was the population surrounding lakes: 23 800 persons living in a 5 km buffer around a lake were equivalent to the presence of a marina.

The travel times between jurisdictions and recipient lakes had a huge effect on the expected numbers of travelling boaters. Boater counts decreased in cubic order of the travel time.

*Route choice model*. The fitted route choice model suggests that boaters have a strong preference for the shortest route. According to the model, an alternative route only 10% longer than the shortest route attracts only half as many agents. The parameters for the best-fitting route choice model are displayed in table 2.

*Temporal pattern model*. Our fitted traffic pattern model has the traffic peak at 14.00 with 95% confidence interval [13.48, 14.20]. For the scale parameter, which determines how sharp the traffic peak is, we obtained a value of 1.34 with confidence interval [1.11,1.56]. This implies that the boater traffic density during midday is about 15 times as high as at night. The probability density function of the temporal pattern model is plotted in figure 4.

*Compliance model*. The estimated proportion of boaters participating in the survey was 80%. Out of these boaters, 93% delivered consistent and complete data. The overall rate of boaters providing useful information was thus 74.4%.

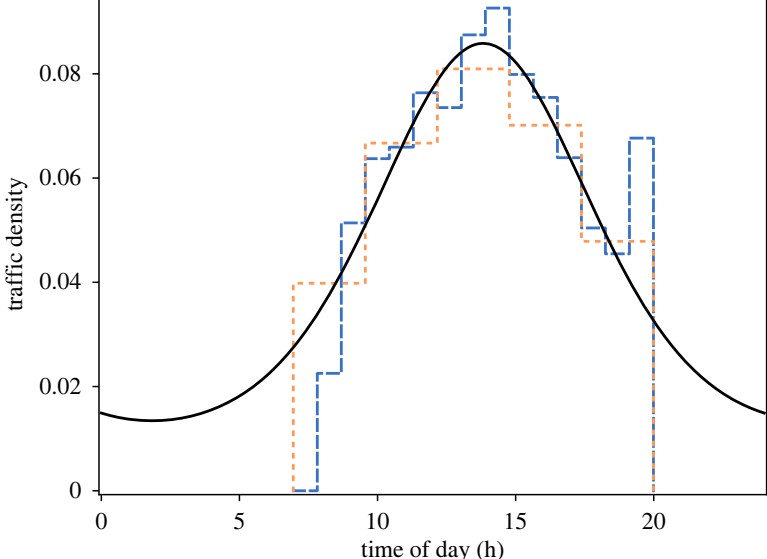

**Figure 4.** Traffic profile. The solid black line depicts the fitted von Mises probability density function modelling the traffic density. The dashed lines depict fitted step function distributions with 5 (short orange dashes) and 15 (long blue dashes) intervals. The step function distributions do not have a predetermined shape such as the von Mises distribution; nonetheless, they match the shape of the von Mises distribution. The step function distributions could not be fitted to night traffic due to missing overnight survey data. Refer to electronic supplementary material, appendix F, for details on how the figure was created.

### 4.2.2. Propagule transport

*Donor regions.* According to our model, most of the external boaters driving to BC come from Alberta (71%) and Washington (19%). However, we did not consider these jurisdictions as potential propagule donors. The most significant sources of high-risk boaters were Saskatchewan (4.3% of the total inflow) and Manitoba (1%). Note that we treated Saskatchewan as a *potential* donor of dreissenid mussels even though no dreissenid mussels have been found in the province to date (see §4.1). In total, the Canadian provinces were contributing more than three times as many high-risk boaters as the American states. In figure 5, we depict the respective contributions of the potential donor regions.

The inflow of high-risk boaters concentrates on few lakes in BC. The nine most-frequented lakes receive 50% of the total high-risk boater pressure; the top 157 lakes receive 90% of the total high-risk boater pressure. The lakes attracting most high-risk boaters were Okanagan Lake (received 17% of all high-risk boaters), Kootenay Lake (7.5%) and Shuswap Lake (6%). These lakes are large and located in the populated southern part of BC. See figure 6 for a map showing the high-risk boater arrivals for the British Columbian lakes.

*Most frequented roads.* In figure 7, the high-risk boater traffic is mapped onto the highway network of BC. The traffic concentrates on a small set of major roads accommodating traffic to clusters of many or highly attractive lakes. Thereby, the roads crossing the eastern border of BC, in particular the Trans-Canada Highway, have the highest boater counts.

### 4.2.3. Validation results and accuracy

The confidence intervals for the model parameters are provided in tables 1 and 2. The normalized predicted and observed boater count values along with predicted 95% confidence intervals are plotted in figure 8; 78% of the count values from the dataset used to fit the model and 67% of the count values from the validation set were within the predicted 95% confidence interval. The $R^2$-value was 0.79 for the data used to fit the model and 0.73 for the validation data. The pseudo $R^2$-value for the overall model was 0.60 for the data used to fit the model and 0.62 for the validation data.

## 5. Discussion

We presented a hybrid gravity, route choice, temporal pattern and compliance model to assess traffic flows in realistic continent-sized road networks. The hybrid model can be used to estimate the agent

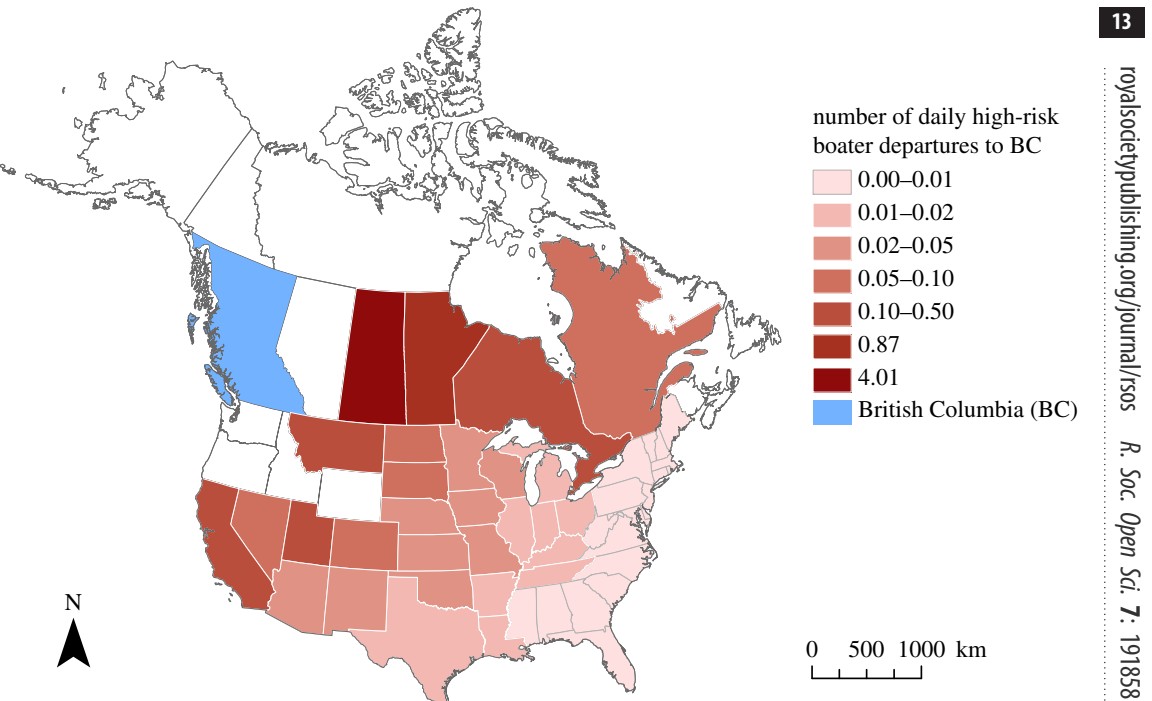

**Figure 5.** Potential donor regions of dreissenid mussels. The red shading depicts how many boaters are estimated to drive from the jurisdictions to BC each day.

outflow of donor regions, the agent traffic volume on roads and the arrival counts of agents at recipient regions. We provided both a general framework for building traffic models based on field traffic survey data as well as a set of directly applicable submodels. We demonstrated the applicability of our approach by studying the inflow of potentially mussel-infested boats to the Canadian province BC.

Combining a gravity, route choice, temporal pattern and compliance model has two major advantages: data can be collected and used more efficiently, and the combined models yield more information than the submodels individually. First, data collected at few locations in the road network can be used to draw inference on the traffic between many origin–destination pairs at once. This makes it possible to assess traffic even in continent-scale road networks. Second, neither a gravity model nor a route choice model alone could provide estimates of how many agents travel along a specific road. A model predicting *how many* agents travel is required as much as a model predicting *where* these agents drive. Thus, our combined approach is more powerful than sequential individual modelling efforts.

## 5.1. Data sources for gravity models

Various data sources have been used to fit gravity models in ecology. However, as will become apparent below, these data sources have considerable limitations in many scenarios.

Most studies in ecology are based on data gathered in mail-out surveys (e.g. [11,12,17]). With data collected via this method, the gravity model could be fitted separately from the other submodels, whereby known methods could be applied (e.g. [12]). To fit the route choice model, data on route choice would be needed in addition to data on boater origins and destinations.

Fitting the submodels individually to mail-out survey data could potentially improve the fit of the hybrid model if enough data are available. However, despite this potential advantage and though mail-out surveys are often the easiest method to gather data to parametrize gravity models, these surveys are subject to significant sampling error, in particular if only few of the surveyed potential travellers ever start a trip. For example, only few boat owners in the Canadian province Ontario will ever take their boat on a 30 h trip to the province BC. Consequently, if the boater traffic between these provinces shall be studied, identifying and contacting the corresponding boaters for a survey will be challenging, and only little data will be available to fit the model. In addition to this issue, mail-out surveys can only yield relative traffic estimates unless further data are available to calibrate the model, and mail-out survey data are not well suited to distinguish the stochasticity inherent to the system from model inaccuracy, unless the survey is repeated, which is rarely done.

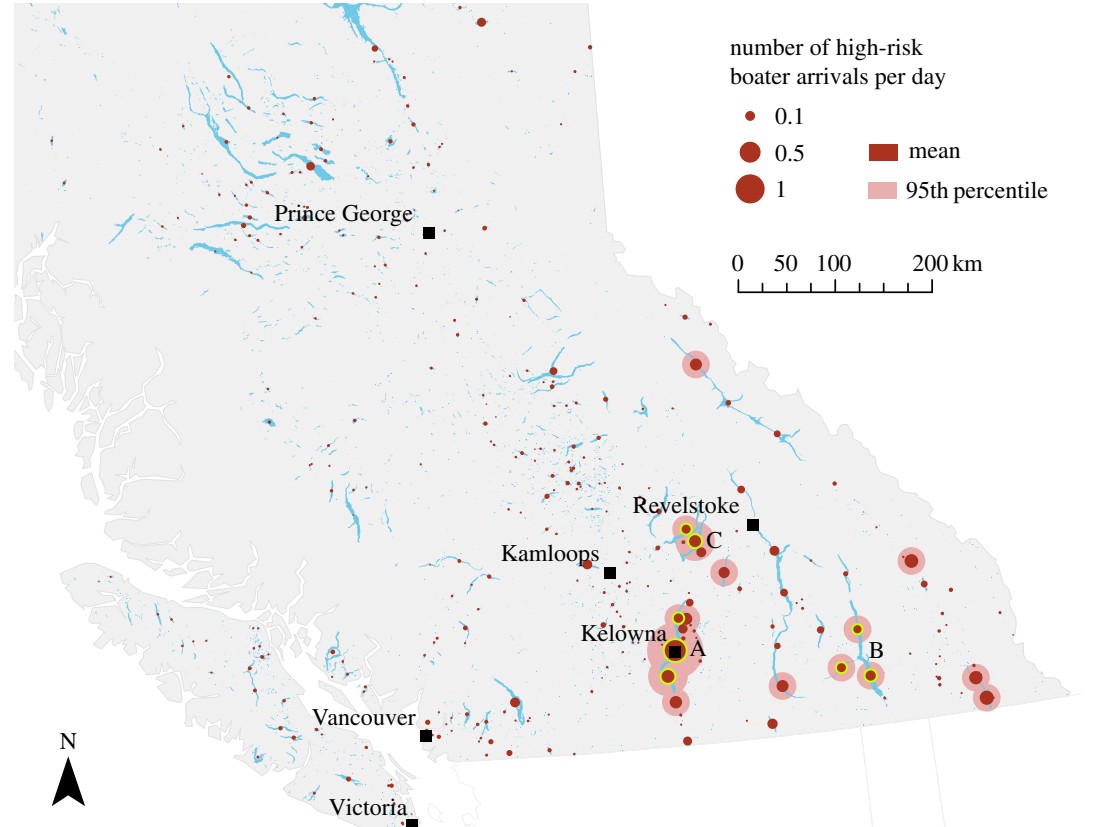

**Figure 6.** Daily arrivals of potentially infested boats at 4700 British Columbian lakes. The sizes of the red circles correspond to the respective arrival counts (dark red: expected number; light red: 95th percentile of arrival counts). Subsections of large lakes are treated as separate lakes to allow for a higher spatial resolution. The letter labels correspond to the three lakes with the highest boater inflow (summed over all subsections, highlighted yellow): A, Okanagan Lake; B, Kootenay Lake; C, Shuswap Lake.

As an alternative to using mail-out survey data, some studies in ecology consider survey data collected at a small sample of origin or destination locations to fit gravity models [37]. Similar to mail-out surveys, these data are prone to sampling error. In addition, special care has to be taken to ensure that the sample of origins or destinations is representative. Otherwise, the data will lead to biased estimates.

In some rare cases, traffic data can be obtained from booking systems at the destination locations [38]. These data sources are among the best possible foundations for fitting gravity models. However, data from booking systems are often not available, especially in large-scale systems, in which each destination may cover a large area.

Lastly, some studies in invasion ecology combine a gravity model with an establishment model, which maps the output of the gravity model to invasion probabilities. Then, the joint model is fitted to data of the temporal progression of the considered invasion [39–41]. This approach can be taken only if the invasion has already progressed sufficiently far and the temporal progression of the invasion is known. Furthermore, this method may not yield concrete estimates of the traffic flows, because some traffic-related parameters may remain unidentifiable if gravity and establishment model are fitted simultaneously [40]. Consequently, a combined gravity and establishment model is useful only in specific cases.

Due to the difficulties associated with using the commonly used data sources to fit gravity models, roadside surveys may often be the most practicable, precise and cost-effective data source for traffic estimates. The hybrid gravity and route choice model makes these data available for fitting gravity models.

Though the presented model for the transport of propagules or pathogens in large-scale systems is new, other studies have considered large-scale invasions before [37,41]. These studies reduce the need for survey data by making strong assumptions on the drivers of repulsiveness and attractiveness. However, the models may suffer from inaccuracy, since large parts of the models are fitted without

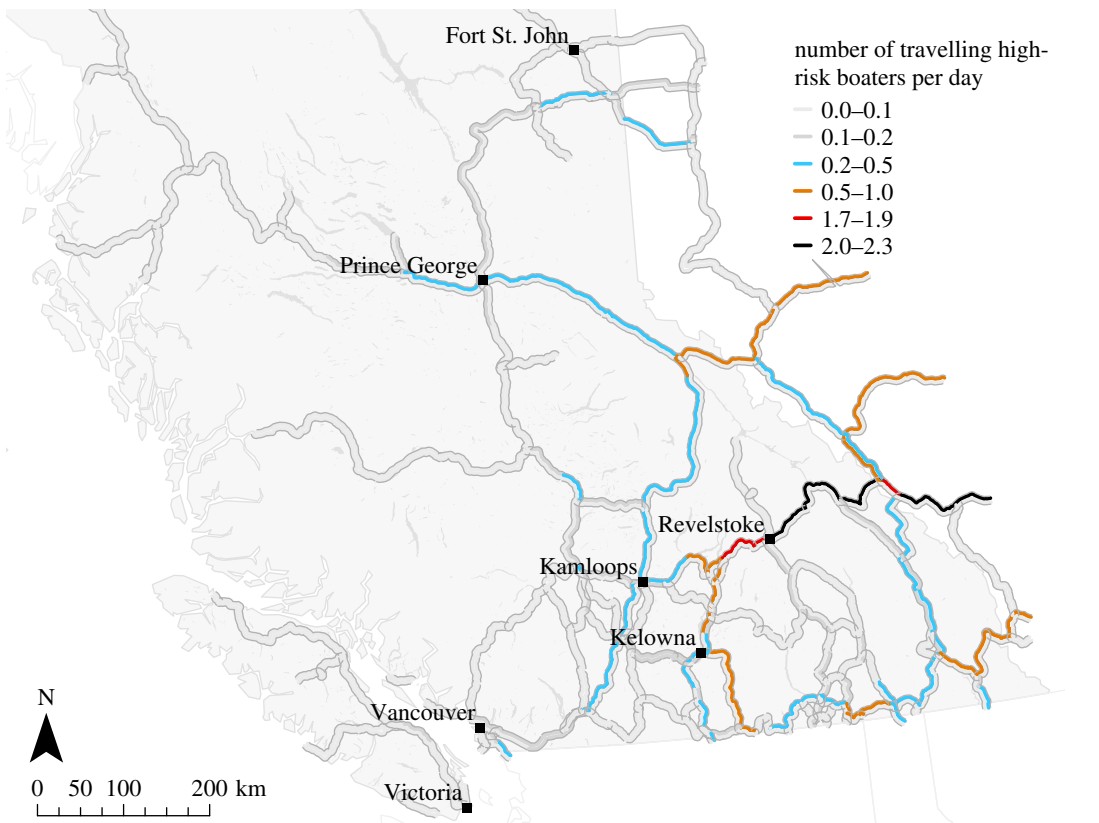

**Figure 7.** Traffic of potentially infested boats along major British Columbian roads. The colours correspond to the expected daily numbers of travelling boaters. The roads' lanes are coloured separately to depict the traffic in different driving directions (assuming right-hand traffic).

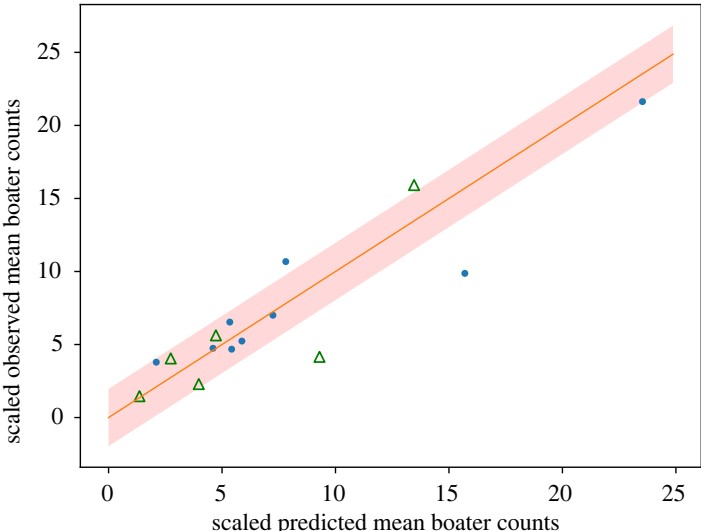

**Figure 8.** Predicted and observed mean boater counts at survey locations. The shaded area is the 95% confidence interval within which the observations would be expected according to our model. 75% of the data lie within the 95% confidence interval. Consequently, the model fits the data well but not perfectly. The blue dots correspond to data used to fit the model, whereas the green hollow triangles correspond to validation data. The solid line shows where predicted and observed values would be equal. All values have been scaled by their respective predicted standard deviation. As a consequence, a one-unit deviation between a predicted and observed value corresponds to a difference of one predicted standard deviation.

survey data. In fact, errors resulting from the additional assumptions cannot even be measured, because no data are available to validate the models rigorously. Furthermore, the added assumptions also decrease model portability [12]. Thus, the hybrid model, fitted with actual survey data, has strong advantages over earlier large-scale models.

## 5.2. Model validation and accuracy

The $R^2$-value we determined for the variance-corrected boater counts at survey locations (0.73 for the validation data) exceeded similar metrics obtained for gravity models previously. For example, Chivers & Leung [17] obtained a value of 0.58 for a model estimating the boater inflow to 781 lakes in Ontario, Canada, and Drake and Mandrak [11] obtained a value of 0.48 for counts of angler trips between 207 origins and 407 destinations in Ontario.

Note, however, that $R^2$-values depend on the measured quantity and the considered system. For example, it is easier to obtain high $R^2$-values for aggregated quantities, such as traveller counts at roads, than for separate quantities, such as trip counts for individual origin–destination pairs. Similarly, high $R^2$-values are easier to obtain in medium-scale systems with little spatial variation, such as in Ontario, than in large-scale systems with high spatial heterogeneity, such as in the continental-scale system we considered. Another factor making direct comparison of the reported quality metrics difficult is that slightly different approaches were used to compute the values. Note that the rescaling step we applied *decreased* the $R^2$-value as compared with the value for raw data. Despite these difficulties in comparing the $R^2$-values, we may conclude that our hybrid model applied to a large-scale system had comparable predictive quality to similar models for medium-scale systems.

In contrast to $R^2$, which is computed with respect to a specific quantity of interest, Nagelkerke's pseudo $R^2$ assesses how well the entire model fits the data. However, though the pseudo $R^2$ reduces to the classical $R^2$ in linear regression, pseudo $R^2$ may be difficult to interpret quantitatively, because it is a relative measure comparing the best-fitting model with a null model. As a consequence, the pseudo $R^2$ depends on the choice of the null model and could be high even if both the null model and the best-fitting model were misspecified. Nonetheless, the relatively high pseudo $R^2$-value we obtained (0.62 for the validation data) indicates that the submodels we used improve the model fit substantially compared with non-informative alternatives.

$R^2$-values yield insights into how well models predict specific quantities of interest. A deterministic model is deemed optimal if it predicts observed data perfectly, and $R^2$ measures how close a model is to this ideal. However, in stochastic modelling, the quantities of interest are deemed inherently stochastic, i.e. not exactly predictable. Consequently, the goal is to predict the *distribution* of the values accurately. A stochastic model could thus have a large $R^2$-value despite modelling the considered process poorly or, if the regarded system is highly stochastic, have a small $R^2$-value despite matching the distribution perfectly. Therefore, $R^2$-values do not suffice to validate stochastic models.

To validate our model further, we compared the differences between predicted and observed values to the deviations predicted by the model. Despite the relatively high $R^2$-values we obtained, the differences between observations and predictions were larger than expected. This indicates that while major components of the boater traffic were captured by the model, some mechanisms might still be missing. A thorough analysis of the gravity model components (electronic supplementary material, appendix F) suggested that the highest potential of improvement is in enhancing the model's ability to distinguish between attractive and unattractive lakes. This, however, would require further data about lakes. In addition to comparing predicted and observed values, we confirmed that the negative binomial distribution is indeed suited to model boater counts and that our model does not yield biased predictions (see electronic supplementary material, appendix F).

The computed parameter confidence intervals revealed an estimability problem with respect to the parameters modelling the increased attractiveness of lakes surrounded by a high population. Closer investigation of the issue showed that a simplified model fits the data reasonably well, too, causing one of the parameters to become potentially obsolete (see electronic supplementary material, appendix G). Both the full and the simplified model yield similar traffic estimates, because the additional parameter affects only lakes in highly populated areas, which are rare in BC.

Though the discussed estimability issue does not affect the traffic estimates for BC, additional analysis will be necessary before the fitted model can be applied to estimate boater traffic in systems with many lakes in highly populated areas. In such systems, the model would need to be refitted, which would yield more precise estimates for the parameters in question.

Besides the mentioned estimability issue and the known difficulty to estimate traffic on inadmissible routes, the confidence intervals for the model parameters were relatively narrow. This suggests that the remaining parameters are well estimable and that potential multicollinearity in covariates is not of concern.

An additional approach to detect potential sources of prediction error could be to conduct a sensitivity analysis. If a model is highly sensitive to a parameter, model predictions may be subject to considerable error unless this parameter is known precisely. This approach is of particular use when

model parameters are estimated separately from one another because it can pinpoint where additional effort should go to reduce parameter uncertainty. However, in the context of estimating parameters simultaneously via maximum-likelihood estimation, as we have done, parameters that have a high impact on predictions often also have a high impact on the likelihood function. As a result, these parameters typically have narrow confidence intervals and are thus known with high precision. This decreases the risk that these parameters act as major error sources, and thus a sensitivity analysis may not be suited to identify the most significant error sources in our case. For this reason, we did not conduct a sensitivity analysis in this study.

## 5.3. Applications

The primary purpose of the hybrid model presented in this paper is to study the traffic of agents potentially carrying propagules or pathogens. If the travel behaviour of these agents is known, early detection and control actions can be implemented more effectively. Thus, the hybrid model can help managers to control invasions and infectious diseases.

First, the hybrid model can facilitate early detection of invasions and infections by providing estimates of the number of potentially infested agents arriving at susceptible locations. These estimates are a valuable proxy for propagule or pathogen pressure and have been used to estimate invasion or infection risk [16,38,39]. These risk estimates, in turn, could be applied to allocate early detection effort and rapidly deploy resources to the locations that are threatened most.

Second, the hybrid model's estimates of agent traffic along roads can be used to decrease invasion or infection risk before infestations occur. For example, invasive species managers in BC set up watercraft inspection stations on roads to detect and treat mussel-infested trailed watercraft. Since most long-distance traffic concentrates on a small number of roads, it is much more efficient to apply such control measures on intermediate roads rather than at the access points of susceptible locations. Our hybrid model could be used to facilitate the choice of optimal control locations.

When using the hybrid model to find optimal control locations, it is helpful that the model not only estimates the agent traffic at all considered roads but also predicts how control applied at one road affects the remaining propagule or pathogen flow at other roads. As a consequence, the hybrid model has the potential to aid management much better than simple traffic measurements on roads, the momentarily common method to identify good control locations.

Besides facilitating management of invasions and infectious diseases, the hybrid model could also lead to a more comprehensive general understanding of human-aided dispersal of species. As the hybrid model focuses on agents that have the potential to carry several invasive species, it would be possible to investigate the dispersal of multiple species with a single modelling effort. The option to incorporate many origin–destination pairs with relatively low survey effort would allow comprehensive studies. This could help ecologists to gain a deeper understanding of the dispersal of both native and invasive species and to assess the impact of road traffic on ecosystems.

## 5.4. Limitations

Since the hybrid model involves four submodels for specific agent decisions, it has a considerable level of complexity, which we aimed to reduce by using simple submodels. As a consequence, some of the proposed submodels may seem unrealistic. Nonetheless, we argue that the proposed models provide valuable insights despite their limitations.

First, we assumed that the compliance of agents is independent of when and where the survey is conducted and who is surveyed. However, particularly the survey location can play a major role for the compliance of agents. For example, more agents may participate in the survey at a boarder crossing, where all travellers have to stop. However, we chose our survey locations carefully with proper signage, and compliance was mandatory. This decreases the variations of the compliance rates.

Second, we accounted for temporal traffic variations with a simple two-parameter model. Thereby, we ignored weekly and seasonal traffic patterns and assumed that the temporal traffic distribution is independent of the sampling location. In reality, traffic is likely to follow more complex patterns. However, even if the fitted temporal traffic distribution does not match the data perfectly, the introduced error will be small, unless the model is very far from the real traffic pattern. Furthermore, the overdispersion resulting from not properly modelled weekly and seasonal traffic patterns is phenomenologically accounted for with the negative binomial distribution. Therefore, our simple

temporal traffic pattern will yield generally accurate estimates, even though estimates resulting from a more sophisticated model could be more precise.

Third, we assumed that agents base their route choices solely on expected travel time, and we ignored potential issues arising from overlapping admissible routes [42]. In addition, our noise traffic model, accounting for agents travelling along inadmissible paths, allows unrealistic disconnected routes. All these issues could be resolved by using more sophisticated submodels. However, our simple noise traffic model affects only 5% of the traffic flow, because most boaters are assumed to choose admissible routes. Modelling routing decisions more realistically could make further data necessary, and the model fit would become computationally harder. We believe that our route choice model constitutes a good first approximation of routing decisions.

Fourth, we made several approximations via independence assumptions. These assumptions decrease the meaningfulness of confidence intervals and model selection criteria (see electronic supplementary material, appendix E). Nonetheless, parameter estimates remain unbiased [43], while the gain of computational efficiency resulting from the independence assumptions is considerable. In fact, accounting for all potential dependencies could make the model fit computationally infeasible. Therefore, the independence assumptions may be a necessary concession to computational efficiency.

The precision of the hybrid model is strongly dependent on how well the available covariates describe attractiveness and repulsiveness of origins and destinations. Due to this limitation, the differences between predictions and observations were larger than expected for our boater traffic model (see electronic supplementary material, appendix F). However, model accuracy is always dependent on the explanatory power of the used data. Therefore, it is unlikely that a different model based on the same data would yield significantly more precise estimates.

Note that though a more precise model might be desirable, the rigorous model validation that revealed our model's inaccuracies would have been hardly possible without the comprehensive survey data made available through the hybrid approach. For example, mail-out surveys are typically designed as cross-sectional studies. Solely based on these data, it is difficult to determine whether differences between model predictions and observations are due to random processes or due to a poorly fitting model. A longitudinal study, such as repeated collection of count data at road sides, is required to discern between prediction error and stochasticity inherent to the modelled system. Gravity models purely based on cross-sectional data may therefore yield misleading results.

Given that existing models could not be validated as rigorously as ours and considering the moderately high $R^2$ values we obtained, we do not have evidence that our hybrid model of boater traffic is less accurate than similar models presented earlier. Quite the contrary, the hybrid model could make a contribution to reveal hidden shortcomings of commonly used models.

When applying the hybrid model to new systems, caution is necessary if the studied agents have an interest to avoid surveys, e.g. because infested agents are fined. In such scenarios, agents may seek routes bypassing survey locations. This would bias the traffic estimates. However, as for our study, it is unlikely that the presence of survey stations impacted boaters' routing decisions significantly, because no fines were imposed on compliant boaters. Furthermore, the road network in BC is relatively sparse, so that detours bypassing survey stations typically take more time than the survey itself. In denser road networks, nonetheless, the impact of survey stations on routing decisions may need to be considered.

Though we considered an extensive set of potential boater destinations, we did not consider rivers. Due to the water current, rivers have a lower risk of being invaded by dreissenid mussels [44]. Nonetheless, some river sections may be at risk. Including rivers as boater destinations is challenging, because it is difficult to identify the specific locations where boaters access the rivers. This was not possible with our dataset, and we did not consider boaters travelling to rivers. However, future studies may consider rivers in addition to lakes.

## 5.5. Future directions

A strength of our approach is in its flexibility. The model fitting techniques that we presented in this paper remain applicable if submodels are exchanged or added. Therefore, we hope that future research will build on this study and develop adjusted and refined submodels to tackle different problems in invasion ecology and epidemiology.

The hybrid model gives insight into the traffic of potential invasive species vectors. As such it provides valuable insights into the progression of invasions. A comprehensive invasion model, however, would need to consider the density of propagules in the donor regions, the mortality of

propagules on the journey, and the habitat suitability at the recipient regions. It will be a valuable task for future research to extend the hybrid model to include the additional invasion stages.

The increased amount of survey data made usable by our approach can also lead to new methodological results. The newly available data may allow modellers to incorporate more covariates in gravity models and use more effective methods to draw inference from the covariates. For example, machine learning techniques could be used to compute repulsiveness and attractiveness of origin and destination locations more accurately. This could lead to traffic models with a new level of predictive quality.

Additional data could be used to fit more sophisticated models for compliance, temporal traffic patterns, and route choice. Compliance rates could be estimated for each survey location independently. Furthermore, the conditional likelihood method presented in this paper could easily be extended to fit a temporal traffic pattern model accounting for weekly and seasonal cycles. Alternatively, a gravity model with a temporally variable mean could be used. Route choice probabilities could be computed based on a variety of route characteristics, such as the scenery or the number of sights along a route (e.g. [45]). With such improvements, the model could become more accurate.

New and more precise ways of fitting the gravity model could be developed if cell phone tracking data of agents are available. Such data could not only yield precise measures of relative count data but also be used to fit a more realistic route choice model, potentially even without computing admissible routes first [46]. With such improvements, agent traffic could be predicted and understood more precisely.

The results on agent flows computed with the techniques presented in this paper open new possibilities for optimizing invasion and disease control measures. If agent traffic flows are known, methods from optimal control theory could be used to improve control strategies and determine locations where control measures are most effective. Furthermore, the model could be used to study the potential impact of a progressed invasion or changed boater behaviour on management. Consequently, this study provides the prerequisites for a number of highly relevant management problems.

Ethics. This article does not present research with ethical considerations.

Data accessibility. Datasets compiled for this research are available as electronic electronic supplementary material. The sources of the original data used to generate the compiled datasets are provided in electronic supplementary material, appendix C. The code used in this study is publicly available at http://vemomoto.github.io and has been archived within the Zenodo repository: https://doi.org/10.5281/zenodo.3764130.

Authors' contributions. All authors conceived the project; S.M.F. and M.A.L. conceived the methods. M.B. and L.-M.H. provided the survey data; M.B. and S.M.F. prepared the data for the analysis. S.M.F. conducted the mathematical analysis, implemented the model and wrote the manuscript. All authors revised the manuscript.

Competing interests. We declare we have no competing interests.

Funding. S.M.F. is thankful for the funding received from the Canadian Aquatic Invasive Species Network; M.A.L. gratefully acknowledges an NSERC Discovery Grant and Canada Research Chair.

Acknowledgements. The authors would like to give thanks to the BC Ministry of Environment and Climate Change Strategy staff of the BC Invasive Mussel Defence Program, who conducted the survey this study is based on. Furthermore, the authors thank the members of the Lewis Research Group at the University of Alberta for helpful feedback and discussions.

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
