## [Reviewer comments · Royal Society Open Science]

Review History

RSOS-191858.R0 (Original submission)

Review form: Reviewer 1

Is the manuscript scientifically sound in its present form?

Yes

Are the interpretations and conclusions justified by the results?

No

Is the language acceptable?

Yes

Do you have any ethical concerns with this paper?

No

Have you any concerns about statistical analyses in this paper?

Yes

Recommendation?

Major revision is needed (please make suggestions in comments)

Comments to the Author(s)

Manuscript Title:

A hybrid gravity and route choice model to assess vector traffic in large-scale road networks

Summary:

In this paper, the authors present a novel model to characterize the movement of vectors across road networks that combines both gravity model and route choice components and can be applied at large scales in a computationally feasible way. The model is a four-part hierarchical framework that explicitly models compliance and patterns of survey effort in combination with route choice and road network models. This approach allows for the discrimination between factors influencing source and destination attractiveness and repulsiveness from factors affecting the choice of movement paths between sites, and has many potentially useful applications for predicting the spread of invasive species and effectively defining priority sites for surveillance and control. The authors use zebra mussel invasion in British Columbia as a case study to demonstrate the applicability of and validate their approach, and find it is moderately predictive and that most parameters are identifiable.

General comments:

Firstly, I believe that keeping in mind the principles of RSOS, the authors should provide at least some example code for this model. This would not only allow reproducibility and for faster implementation by others, but would provide additional evidence of the model's fast runtime without the reader having extensive knowledge of runtime analysis. Given the broad readership of the journal, and the fact that many of the invasion biologists who would be interested in this approach do not have the mathematical training necessary to replicate the analysis from the equations provided, the inclusion of code would be necessary to render this methodology broadly accessible.

The main body of the paper is written clearly and succinctly. I admire the authors' commitment to maximizing the mechanistic realism of the vector movement process through their hierarchical approach, and while I do think the data requirements are quite high in order to replicate the case study, this is likely not the only case for which it is applicable, though I would have appreciated a discussion of how the model could be modified for cases where such a thorough survey protocol was not possible (e.g. how can mail-out survey data be incorporated?).

The paper's appeal could be much more broad if more of the case study results were included in the main text, and the inclusion of validation results in the main text would provide a more honest picture of the demonstrated strength of the methodology. While I think the novelty of this method is sufficient for this work to be highly important in spite of only moderate demonstrable predictive power, I think that the results should be presented more transparently in text. The inclusion of only fitted data in in-text figures masks the predictive ability of the model, which is only visualized through predicted vs. observed plots in the supplement. In addition, the authors refrain from referring to any quantitative metrics of fit. I understand that the model does not conform to the assumptions of traditional R² metrics, but given that the alternative is visual inspection, I would argue that readers will interpret plots by visually estimating R² regardless of whether it is appropriate (e.g. perhaps the author's could provide a Pseudo-R² metric for a negative binomial GLM like Nagergelke). Further, while I understand the full scope of validation presented in the supplement cannot be applied to other gravity models, I do think that some comparison of predictive ability to previous work should be made (how strong are predictions of source/destination pairs?) in order to assess the performance of this approach relative to other methods.

Finally, I cannot fully evaluate the detailed mathematical proofs within the supplement, and would recommend the editor seek out a reviewer with the appropriate training in order to fully assess their validity.

Specific comments (in all cases I use the innermost line numbers):

Abstract:

I recommend including some quantitative metric of model fit for the case study here (like a Pseudo R²).

Introduction:

lines 88-91: isn't this easily circumvented by only inquiring about historical trips?

Model:

Overall, I find the schematic figures very helpful and the approach clear to follow.

Eqn 1 - c should be defined in text

Lines 179-181: What about factors that have both a main effect and interaction?

Lines 204-207: I am not sure I follow this logic - why would the rarity of total agents of concern affect whether these agents travel independently?

Lines 217-219: Is there any evidence in the literature that most travellers make at the most one intermediate stop?

Model Fit:

I believe that some mention of the software used for fitting the model is necessary in the main text.

Application:

Lines 323-327: Was there a specific boat size cutoff used?

Lines 358-362: How correlated were the predictor variables tested across submodels? How well does the model distinguish highly correlated predictors? A correlation matrix or even some discussion on the relative identifiability as it relates to correlation strength would be useful in an appendix.

Figure 4: Without the raw data, this figure does not seem very informative.

Lines 376-382: I think the inclusion of the parameter estimates in Appendix F, or a reduced version of it, is worth including in the main text.

Line 386: While this is more relevant to a subsequent model of invasion risk, I feel some discussion is necessary about how donor regions far from recipients will have higher propagule mortality and thus lower risk.

Figure 6 - It is unclear which circles correspond with A,B,C as there are multiple access points - perhaps colour code A B and C lakes to ease interpretation. Further, is there anyway to visualize uncertainty in these plots? perhaps using transparent inner and outer circles?

Figure 7 - Similarly, a depiction of uncertainty would be useful here. Perhaps even as CI values on the map itself

Discussion:

Line 411- perhaps I'm unfamiliar with this terminology, but I would suggest 'recipient regions' to better match your wording of 'donor regions'

Lines 415-417 - at times I find that the paper reads more like a demonstration of the utility of surveying along roads rather than the model itself. I think this could be improved by an additional discussion of how other sources of data that may already exist could also be included (e.g. mail-out surveys or surveys at lakes), and if partial information e.g. from booking systems can be somehow incorporated in cases where such a thorough survey is not possible.

Line 495 - remove 'they'

Line 497 - again, this is more of an advantage of intensive surveying rather than the methodology presented, per se.

Line 500 - Is there a chance that the implementation of required stopping points along roads influenced traffic patterns?

Line 510 - Is there a way to estimate what fraction of included potential routes are unrealistic?

Supplement:

Line 26: Is there any potential to create gradations of route admissibility in order to limit the number of clearly impossible routes?

Line 33: How important are multiple trips by the same agent? Does a large fraction of multiple trips affect the assumptions of independence?

Line 79: How might this varying coarseness impact the management implications of this approach? e.g. I would think it becomes less clear how to target non-BC sites for surveillance/control measures.

Lines 81-86: I think this information is worth moving to main text.

Line 99: affect → effect

Line 103: Again, prefer 'recipient regions'

Line 104: Perhaps I'm misunderstanding, but I think the word 'not' is superfluous here

Line 116-117: As mentioned above, I think facilitating implementation is of key importance in keeping with the principles of Open Science, and that details and code should be made available.

Line 236: I would change the notation d to a different letter so that it is not confused with distance in the gravity model equations in the main text.

Eqn A5: What is the meaning of the tilde in these equations?

Lines 464-465: I think this information should be included in the methods of the main text.

Line 491: loose → lose

Line 494: I would recommend predictive performance on a validation set as a better alternative to AIC (e.g. choosing the model that minimizes RMSE).

Line 508: I would be careful with this justification, as the mechanistic separation of traffic flow and route choice is one of the arguments for building the model in the introduction, and no clear advantage has been shown for this model's ability to predict the correct propagule pressure relative to less mechanistic alternatives.

Lines 522-528: I think this is worth moving to the main text.

Line 535: The separability of predictive factors is still useful in order to extrapolate the model predictions across time and space. How sensitive would the model be to changes in correlation structure of the modelled predictors?

Table A3: I would recommend including this table in the main text, and would recommend centring and scaling the predictor variables so that their relative magnitudes are meaningful.

Line 564: I would like to see the raw data plotted on Figure 4.

Line 962: This is where I think RMSE and Pseudo-R² metrics would be advantageous to present. Regardless of their validity, the reader is going to compare the points to a 1-to-1 line in a way that approximates R² anyway. It also facilitates the interpretation of the relative amount of error in each component of the model, and RMSE could be used for model comparison as an alternative to AIC given that AIC assumptions are also not met.

Lines 1007-1008: This result is particularly troubling, as the source and destination of vectors is key for predicting the spread of invasions. This is where a comparison between this model's predictive power and that of previous gravity models would strengthen the paper.

Figure A3: For transparency, I think this needs to be in the main text. Do the author's have any ideas on the causes of major outliers?

Lines 1080-1086: I think this paragraph would also fit well in the discussion after moving more of the case study methods and results to the main text.

Line 1090: issued –>issues

Line 1092: But would this make the model's predictive power sensitive to fluctuations in population size over time? Can this model effectively forecast vector movement with changing human population projections?

Review form: Reviewer 2 (Bernd Blasius)

Is the manuscript scientifically sound in its present form?

Yes

Are the interpretations and conclusions justified by the results?

Yes

Is the language acceptable?

Yes

Do you have any ethical concerns with this paper?

No

Have you any concerns about statistical analyses in this paper?

No

Recommendation?

Accept with minor revision (please list in comments)

Comments to the Author(s)

The study follows two goals, i) to present a hybrid gravity and route choice model for estimating traffic in a transportation network, and ii) to apply this model for investigating the potential invasion of Zebra and Quagga mussels into British Columbia. This is an interesting and stimulating study. The proposed hybrid model is an interesting contribution to traffic modelling. The model is also well presented and the supplementary information contains my model details. However, I found some issues, mostly in the application of the model which is only weakly analysed (see comments below).

I would recommend publication of this study when my these issues can be resolved by the authors.

Comments:**Section 2.2 Route change model.**

I had some problems following the rationale behind the model choices here, possibly because I am not familiar with Fischer (2019).

- l1 213-214 "We illustrate this concept.. in Fig.2". The figure does not mention the parameter delta, so how can it illustrate this concept?

- l 216: What is the difference between this rule (using parameter gamma) and the previous rule (using parameter delta). At first glance they seem very similar. Maybe an improved Fig.2 can help here.

- l217: "we focus single-via paths": Please state clearly, what you mean by "focus" in this context. Do you really exclude all paths that contain more than one intermediate step-over location? If yes, can you please give more arguments for this approximation.

- Eq (2): by this equation longer routes automatically are down-regulated. Do we then still need the other extra rules (including delta, gamma, and eta)?

Section 2.3 Temporal patterns

While I can see the elegance to include the temporal dynamics in the general hierarchical framework (Fig. 1), I have a problem to comprehend the relevance of diurnal traffic patterns for this specific application. Do the authors really believe that a resolution to a day-night temporal scale is important to understand the invasion patterns into BC? If yes, can you please give some more arguments?

Note that this is different on whether or not a von-Mises function gives the best description of the temporal pattern (as discussed in detail).

A simple test to demonstrate the prediction-gain by inclusion of the temporal aspect would be to compare model runs with, and without, the temporal components. Would we see differences?

Fig.7, legend "road's lanes are coloured separately to depict traffic in different driving directions": Can you explain this better? I do not understand this sentence and cannot see different directions in the figure.

Results:

After having been introduced to the (rather involved) model, I was somewhat disappointed that the only shown results are three or four geographic pictures. I am missing a discussion of some broader patterns. Some questions along these lines are: Do we have general distance

dependencies in the model outcome? Can we aggregate model outcomes over some variables to learn something about the invasion process?

Furthermore I miss any kind of sensitivity analysis. The Supplementary information contains a wealth of information on the formal aspects of how the fitting was performed, but I miss more simple information on how the model results depend on the many model parameters and model choices?

Decision letter (RSOS-191858.R0)

19-Dec-2019

Dear Mr Fischer,

The editors assigned to your paper ("A hybrid gravity and route choice model to assess vector traffic in large-scale road networks") have now received comments from reviewers. We would like you to revise your paper in accordance with the referee and Associate Editor suggestions which can be found below (not including confidential reports to the Editor). Please note this decision does not guarantee eventual acceptance.

Please submit a copy of your revised paper before 11-Jan-2020. Please note that the revision deadline will expire at 00.00am on this date. If we do not hear from you within this time then it will be assumed that the paper has been withdrawn. In exceptional circumstances, extensions may be possible if agreed with the Editorial Office in advance. We do not allow multiple rounds of revision so we urge you to make every effort to fully address all of the comments at this stage. If deemed necessary by the Editors, your manuscript will be sent back to one or more of the original reviewers for assessment. If the original reviewers are not available, we may invite new reviewers.

- Data accessibility

It is a condition of publication that all supporting data are made available either as supplementary information or preferably in a suitable permanent repository. The data accessibility section should state where the article's supporting data can be accessed. This section

should also include details, where possible of where to access other relevant research materials such as statistical tools, protocols, software etc can be accessed. If the data have been deposited in an external repository this section should list the database, accession number and link to the DOI for all data from the article that have been made publicly available. Data sets that have been deposited in an external repository and have a DOI should also be appropriately cited in the manuscript and included in the reference list.

If you wish to submit your supporting data or code to Dryad (<http://datadryad.org/>), or modify your current submission to dryad, please use the following link:
<http://datadryad.org/submit?journalID=RSOS&manu=RSOS-191858>

- **Competing interests**

- **Authors' contributions**

- **Acknowledgements**

- **Funding statement**

on behalf of Dr Francois Fages (Associate Editor) and Kevin Padian (Subject Editor)
openscience@royalsociety.org

Associate Editor's comments (Dr Francois Fages):

Dear authors,

Your paper has been reviewed by two reviewers who both found your approach interesting but

the analysis of its results on the study cases currently insufficient. You are thus invited to take into consideration the criticisms made in the reviews and revise your manuscript to address them.

Best wishes

Reviewers' Comments to Author:

Reviewer: 1

Comments to the Author(s)

Manuscript Title:

A hybrid gravity and route choice model to assess vector traffic in large-scale road networks

Summary:

In this paper, the authors present a novel model to characterize the movement of vectors across road networks that combines both gravity model and route choice components and can be applied at large scales in a computationally feasible way. The model is a four-part hierarchical framework that explicitly models compliance and patterns of survey effort in combination with route choice and road network models. This approach allows for the discrimination between factors influencing source and destination attractiveness and repulsiveness from factors affecting the choice of movement paths between sites, and has many potentially useful applications for predicting the spread of invasive species and effectively defining priority sites for surveillance and control. The authors use zebra mussel invasion in British Columbia as a case study to demonstrate the applicability of and validate their approach, and find it is moderately predictive and that most parameters are identifiable.

General comments:

Firstly, I believe that keeping in mind the principles of RSOS, the authors should provide at least some example code for this model. This would not only allow reproducibility and for faster implementation by others, but would provide additional evidence of the model's fast runtime without the reader having extensive knowledge of runtime analysis. Given the broad readership of the journal, and the fact that many of the invasion biologists who would be interested in this approach do not have the mathematical training necessary to replicate the analysis from the equations provided, the inclusion of code would be necessary to render this methodology broadly accessible.

The main body of the paper is written clearly and succinctly. I admire the authors' commitment to maximizing the mechanistic realism of the vector movement process through their hierarchical approach, and while I do think the data requirements are quite high in order to replicate the case study, this is likely not the only case for which it is applicable, though I would have appreciated a discussion of how the model could be modified for cases where such a thorough survey protocol was not possible (e.g. how can mail-out survey data be incorporated?).

The paper's appeal could be much more broad if more of the case study results were included in the main text, and the inclusion of validation results in the main text would provide a more honest picture of the demonstrated strength of the methodology. While I think the novelty of this method is sufficient for this work to be highly important in spite of only moderate demonstrable predictive power, I think that the results should be presented more transparently in text. The inclusion of only fitted data in in-text figures masks the predictive ability of the model, which is only visualized through predicted vs. observed plots in the supplement. In addition, the authors refrain from referring to any quantitative metrics of fit. I understand that the model does not conform to the assumptions of traditional R² metrics, but given that the alternative is visual

inspection, I would argue that readers will interpret plots by visually estimating R^2 regardless of whether it is appropriate (e.g. perhaps the author's could provide a Pseudo- R^2 metric for a negative binomial GLM like Nagengelke). Further, while I understand the full scope of validation presented in the supplement cannot be applied to other gravity models, I do think that some comparison of predictive ability to previous work should be made (how strong are predictions of source/destination pairs?) in order to assess the performance of this approach relative to other methods.

Finally, I cannot fully evaluate the detailed mathematical proofs within the supplement, and would recommend the editor seek out a reviewer with the appropriate training in order to fully assess their validity.

Specific comments (in all cases I use the innermost line numbers):

Abstract:

I recommend including some quantitative metric of model fit for the case study here (like a Pseudo R^2).

Introduction:

lines 88-91: isn't this easily circumvented by only inquiring about historical trips?

Model:

Overall, I find the schematic figures very helpful and the approach clear to follow.

Eqn 1 - c should be defined in text

Lines 179-181: What about factors that have both a main effect and interaction?

Lines 204-207: I am not sure I follow this logic - why would the rarity of total agents of concern affect whether these agents travel independently?

Lines 217-219: Is there any evidence in the literature that most travellers make at the most one intermediate stop?

Model Fit:

I believe that some mention of the software used for fitting the model is necessary in the main text.

Application:

Lines 323-327: Was there a specific boat size cutoff used?

Lines 358-362: How correlated were the predictor variables tested across submodels? How well does the model distinguish highly correlated predictors? A correlation matrix or even some discussion on the relative identifiability as it relates to correlation strength would be useful in an appendix.

Figure 4: Without the raw data, this figure does not seem very informative.

Lines 376-382: I think the inclusion of the parameter estimates in Appendix F, or a reduced version of it, is worth including in the main text.

Line 386: While this is more relevant to a subsequent model of invasion risk, I feel some discussion is necessary about how donor regions far from recipients will have higher propagule mortality and thus lower risk.

Figure 6 - It is unclear which circles correspond with A,B,C as there are multiple access points - perhaps colour code A B and C lakes to ease interpretation. Further, is there anyway to visualize uncertainty in these plots? perhaps using transparent inner and outer circles?

Figure 7 - Similarly, a depiction of uncertainty would be useful here. Perhaps even as CI values on the map itself

Discussion:

Line 411- perhaps I'm unfamiliar with this terminology, but I would suggest 'recipient regions' to better match your wording of 'donor regions'

Lines 415-417 - at times I find that the paper reads more like a demonstration of the utility of surveying along roads rather than the model itself. I think this could be improved by an additional discussion of how other sources of data that may already exist could also be included (e.g. mail-out surveys or surveys at lakes), and if partial information e.g. from booking systems can be somehow incorporated in cases where such a thorough survey is not possible.

Line 495 - remove 'they'

Line 497 - again, this is more of an advantage of intensive surveying rather than the methodology presented, per se.

Line 500 - Is there a chance that the implementation of required stopping points along roads influenced traffic patterns?

Line 510 - Is there a way to estimate what fraction of included potential routes are unrealistic?

Supplement:

Line 26: Is there any potential to create gradations of route admissibility in order to limit the number of clearly impossible routes?

Line 33: How important are multiple trips by the same agent? Does a large fraction of multiple trips affect the assumptions of independence?

Line 79: How might this varying coarseness impact the management implications of this approach? e.g. I would think it becomes less clear how to target non-BC sites for surveillance/control measures.

Lines 81-86: I think this information is worth moving to main text.

Line 99: affect -> effect

Line 103: Again, prefer 'recipient regions'

Line 104: Perhaps I'm misunderstanding, but I think the word 'not' is superfluous here

Line 116-117: As mentioned above, I think facilitating implementation is of key importance in keeping with the principles of Open Science, and that details and code should be made available.

Line 236: I would change the notation d to a different letter so that it is not confused with distance in the gravity model equations in the main text.

Eqn A5: What is the meaning of the tilde in these equations?

Lines 464-465: I think this information should be included in the methods of the main text.

Line 491: loose \rightarrow lose

Line 494: I would recommend predictive performance on a validation set as a better alternative to AIC (e.g. choosing the model that minimizes RMSE).

Line 508: I would be careful with this justification, as the mechanistic separation of traffic flow and route choice is one of the arguments for building the model in the introduction, and no clear advantage has been shown for this model's ability to predict the correct propagule pressure relative to less mechanistic alternatives.

Lines 522-528: I think this is worth moving to the main text.

Line 535: The separability of predictive factors is still useful in order to extrapolate the model predictions across time and space. How sensitive would the model be to changes in correlation structure of the modelled predictors?

Table A3: I would recommend including this table in the main text, and would recommend centring and scaling the predictor variables so that their relative magnitudes are meaningful.

Line 564: I would like to see the raw data plotted on Figure 4.

Line 962: This is where I think RMSE and Pseudo-R² metrics would be advantageous to present. Regardless of their validity, the reader is going to compare the points to a 1-to-1 line in a way that approximates R² anyway. It also facilitates the interpretation of the relative amount of error in each component of the model, and RMSE could be used for model comparison as an alternative to AIC given that AIC assumptions are also not met.

Lines 1007-1008: This result is particularly troubling, as the source and destination of vectors is key for predicting the spread of invasions. This is where a comparison between this model's predictive power and that of previous gravity models would strengthen the paper.

Figure A3: For transparency, I think this needs to be in the main text. Do the author's have any ideas on the causes of major outliers?

Lines 1080-1086: I think this paragraph would also fit well in the discussion after moving more of the case study methods and results to the main text.

Line 1090: issued \rightarrow issues

Line 1092: But would this make the model's predictive power sensitive to fluctuations in population size over time? Can this model effectively forecast vector movement with changing human population projections?

Reviewer: 2

Comments to the Author(s)

The study follows two goals, i) to present a hybrid gravity and route choice model for estimating traffic in a transportation network, and ii) to apply this model for investigating the potential

invasion of Zebra and Quagga mussels into British Columbia. This is an interesting and stimulating study. The proposed hybrid model is an interesting contribution to traffic modelling. The model is also well presented and the supplementary information contains my model details. However, I found some issues, mostly in the application of the model which is only weakly analysed (see comments below).

I would recommend publication of this study when my these issues can be resolved by the authors.

Comments:

Section 2.2 Route change model.

I had some problems following the rationale behind the model choices here, possibly because I am not familiar with Fischer (2019).

- ll 213-214 "We illustrate this concept.. in Fig.2". The figure does not mention the parameter delta, so how can it illustrate this concept?

- l 216: What is the difference between this rule (using parameter gamma) and the previous rule (using parameter delta). At first glance they seem very similar. Maybe an improved Fig.2 can help here.

- l217: "we focus single-via paths": Please state clearly, what you mean by "focus" in this context. Do you really exclude all paths that contain more than one intermediate step-over location? If yes, can you please give more arguments for this approximation.

- Eq (2): by this equation longer routes automatically are down-regulated. Do we then still need the other extra rules (including delta, gamma, and eta)?

Section 2.3 Temporal patterns

While I can see the elegance to include the temporal dynamics in the general hierarchical framework (Fig. 1), I have a problem to comprehend the relevance of diurnal traffic patterns for this specific application. Do the authors really believe that a resolution to a day-night temporal scale is important to understand the invasion patterns into BC? If yes, can you please give some more arguments?

Note that this is different on whether or not a von-Mises function gives the best description of the temporal pattern (as discussed in detail). A simple test to demonstrate the prediction-gain by inclusion of the temporal aspect would be to compare model runs with, and without, the temporal components. Would we see differences?

Fig.7, legend "road's lanes are coloured separately to depict traffic in different driving directions": Can you explain this better? I do not understand this sentence and cannot see different directions in the figure.

Results:

After having been introduced to the (rather involved) model, I was somewhat disappointed that the only shown results are three or four geographic pictures. I am missing a discussion of some broader patterns. Some questions along these lines are: Do we have general distance dependencies in the model outcome? Can we aggregate model outcomes over some variables to learn something about the invasion process?

Furthermore I miss any kind of sensitivity analysis. The Supplementary information contains a wealth of information on the formal aspects of how the fitting was performed, but I miss more simple information on how the model results depend on the many model parameters and model choices?

Author's Response to Decision Letter for (RSOS-191858.R0)

See Appendix A.

RSOS-191858.R1 (Revision)

Review form: Reviewer 1

Is the manuscript scientifically sound in its present form?

Yes

Are the interpretations and conclusions justified by the results?

Yes

Is the language acceptable?

Yes

Do you have any ethical concerns with this paper?

No

Have you any concerns about statistical analyses in this paper?

No

Recommendation?

Accept with minor revision (please list in comments)

Comments to the Author(s)

General comments:

I was pleased with the authors' thorough consideration of all of my comments, and believe that the changes that they have made have greatly strengthened their manuscript. I note that I have only examined their responses to my own comments.

I am happy that the authors now provide the code to make their results more transparent and reproducible, that their validation results are more visible, and that they situate their findings within the existing gravity model literature. I also appreciate the increased space given to justifying their particular modelling approach versus conventional approaches.

I have a few specific responses to their changes below:

15. Figure 4: I see now that it is not as simple as plotting a histogram, thank you for the clarification.

38. RMSE discussion:

I respect the authors' choice to perform model selection via AIC, but don't fully agree that this approach ensures the selected model is mechanistically appropriate (from what I've understood of their approach, it would only ensure that the error distributions from which likelihood is computed are appropriate). I also would expect that RMSE could be summed across quantities of interest (analogous to how the likelihood is computed jointly), thereby not requiring the authors to focus on a single quantity upon which to assess model fit. Further, I disagree that RMSE on a validation set is susceptible to overfitting. If some additional noise term is fit to, say, a multiple regression, there is no reason to expect the validation data to have the same spurious correlation with that noise term, so it should not improve RMSE estimates. In any case, I acknowledge that neither metric is perfect, and respect the choice of AIC.

44. I recommend the authors calculate a relative error metric such as relative absolute error (RAE) for the cases in which they believe it is more appropriate than squared error.

Spelling and grammar:

line 258 “needs” not need

line 322: prefer “improve” to “get better at”

line 424: prefer “sharp” to “spiky”

Table 1,2: table column headers seem buggy on my version

lines 684-685: prefer donor and recipient “regions” to maintain consistency

Decision letter (RSOS-191858.R1)

31-Mar-2020

Dear Mr Fischer:

On behalf of the Editors, I am pleased to inform you that your Manuscript RSOS-191858.R1 entitled "A hybrid gravity and route choice model to assess vector traffic in large-scale road networks" has been accepted for publication in Royal Society Open Science subject to minor revision in accordance with the referee suggestions. Please find the referees' comments at the end of this email.

The reviewers and Subject Editor have recommended publication, but also suggest some minor revisions to your manuscript. Therefore, I invite you to respond to the comments and revise your manuscript.

- Ethics statement

- Data accessibility

If you wish to submit your supporting data or code to Dryad (<http://datadryad.org/>), or modify your current submission to dryad, please use the following link:
<http://datadryad.org/submit?journalID=RSOS&manu=RSOS-191858.R1>

- Competing interests

- Authors' contributions

- Acknowledgements

- Funding statement

Because the schedule for publication is very tight, it is a condition of publication that you submit the revised version of your manuscript before 09-Apr-2020. Please note that the revision deadline will expire at 00.00am on this date. If you do not think you will be able to meet this date please let me know immediately.

- 1) A text file of the manuscript (tex, txt, rtf, docx or doc), references, tables (including captions) and figure captions. Do not upload a PDF as your "Main Document".
- 2) A separate electronic file of each figure (EPS or print-quality PDF preferred (either format should be produced directly from original creation package), or original software format)
- 3) Included a 100 word media summary of your paper when requested at submission. Please ensure you have entered correct contact details (email, institution and telephone) in your user account

4) Included the raw data to support the claims made in your paper. You can either include your data as electronic supplementary material or upload to a repository and include the relevant doi within your manuscript

5) All supplementary materials accompanying an accepted article will be treated as in their final form. Note that the Royal Society will neither edit nor typeset supplementary material and it will be hosted as provided. Please ensure that the supplementary material includes the paper details where possible (authors, article title, journal name).

on behalf of Dr Francois Fages (Associate Editor) and Kevin Padian (Subject Editor)
openscience@royalsociety.org

Associate Editor Comments to Author (Dr Francois Fages):

Dear Author

It is my pleasure to accept your paper with minor revision for publication.
Please do take into consideration the last remaining points raised in the review report for producing your final manuscript.

Best regards

Reviewer comments to Author:
Reviewer: 1

Comments to the Author(s)
General comments:

I was pleased with the authors' thorough consideration of all of my comments, and believe that the changes that they have made have greatly strengthened their manuscript. I note that I have only examined their responses to my own comments.

I am happy that the authors now provide the code to make their results more transparent and reproducible, that their validation results are more visible, and that they situate their findings within the existing gravity model literature. I also appreciate the increased space given to justifying their particular modelling approach versus conventional approaches.

I have a few specific responses to their changes below:

15. Figure 4: I see now that it is not as simple as plotting a histogram, thank you for the clarification.

38. RMSE discussion:

I respect the authors' choice to perform model selection via AIC, but don't fully agree that this approach ensures the selected model is mechanistically appropriate (from what I've understood of their approach, it would only ensure that the error distributions from which likelihood is computed are appropriate). I also would expect that RMSE could be summed across quantities of interest (analogous to how the likelihood is computed jointly), thereby not requiring the authors to focus on a single quantity upon which to assess model fit. Further, I disagree that RMSE on a validation set is susceptible to overfitting. If some additional noise term is fit to, say, a multiple regression, there is no reason to expect the validation data to have the same spurious correlation with that noise term, so it should not improve RMSE estimates. In any case, I acknowledge that neither metric is perfect, and respect the choice of AIC.

44. I recommend the authors calculate a relative error metric such as relative absolute error (RAE) for the cases in which they believe it is more appropriate than squared error.

Spelling and grammar:

line 258 "needs" not need

line 322: prefer "improve" to "get better at"

line 424: prefer "sharp" to "spiky"

Table 1,2: table column headers seem buggy on my version

lines 684-685: prefer donor and recipient "regions" to maintain consistency

Author's Response to Decision Letter for (RSOS-191858.R1)

See Appendix B.

Decision letter (RSOS-191858.R2)

Dear Dr Fischer,

It is a pleasure to accept your manuscript entitled "A hybrid gravity and route choice model to assess vector traffic in large-scale road networks" in its current form for publication in Royal Society Open Science. The comments of the reviewer(s) who reviewed your manuscript are included at the foot of this letter.

Please ensure that you send to the editorial office an editable version of your accepted

manuscript, and individual files for each figure and table included in your manuscript. You can send these in a zip folder if more convenient. Failure to provide these files may delay the processing of your proof. You may disregard this request if you have already provided these files to the editorial office.

on behalf of Dr Francois Fages (Associate Editor) and Kevin Padian (Subject Editor)
openscience@royalsociety.org

Appendix A

Associate Editor's comments (Dr Francois Fages):

Dear authors,

Your paper has been reviewed by two reviewers who both found your approach interesting but the analysis of its results on the study cases currently insufficient. You are thus invited to take into consideration the criticisms made in the reviews and revise your manuscript to address them.

Best wishes

Dear Editor,

Thank you for your time and effort in dealing with our manuscript entitled "A Hybrid Gravity and Route Choice Model to Assess Vector Traffic in Large-Scale Road Networks" (RSOS-191858). We appreciate the opportunity to submit a revised version of our manuscript for re-evaluation and would like to thank the reviewers for their constructive criticism and helpful suggestions.

We have thoroughly considered and responded to all reviewer comments and addressed them by changing the manuscript accordingly. In particular, we have carried out a more comprehensive model analysis and included further model validation results. For the few points where we could not satisfy the reviewer comments in full, we provide a careful and in-depth explanation.

In addition to the changes suggested by the reviewers, we have addressed a conceptual shortcoming of the model presented in the original manuscript. The initial model did not account for surveyed boaters who were providing incomplete or inconsistent information. This caused the model to underestimate the boater traffic by 5-10%. We have corrected this issue and adjusted the paper accordingly.

Below please find our detailed answers to the reviewer comments. For easier reference, we have separated and numbered the comments. Our answers are indented and written in blue font below the comments.

We hope that our responses and improvements of the manuscript are suitable to dispel possible concerns and qualify our manuscript for publication in Royal Society Open Science.

Yours sincerely,

Samuel Fischer on behalf of the authors

Reviewer: 1

Manuscript Title:

A hybrid gravity and route choice model to assess vector traffic in large-scale road networks

Summary:

In this paper, the authors present a novel model to characterize the movement of vectors across road networks that combines both gravity model and route choice components and can be applied at large scales in a computationally feasible way. The model is a four-part hierarchical framework that explicitly models compliance

and patterns of survey effort in combination with route choice and road network models. This approach allows for the discrimination between factors influencing source and destination attractiveness and repulsiveness from factors affecting the choice of movement paths between sites, and has many potentially useful applications for predicting the spread of invasive species and effectively defining priority sites for surveillance and control. The authors use zebra mussel invasion in British Columbia as a case study to demonstrate the applicability of and validate their approach, and find it is moderately predictive and that most parameters are identifiable.

General comments:

1. Firstly, I believe that keeping in mind the principles of RSOS, the authors should provide at least some example code for this model. This would not only allow reproducibility and for faster implementation by others, but would provide additional evidence of the model's fast runtime without the reader having extensive knowledge of runtime analysis. Given the broad readership of the journal, and the fact that many of the invasion biologists who would be interested in this approach do not have the mathematical training necessary to replicate the analysis from the equations provided, the inclusion of code would be necessary to render this methodology broadly accessible.
 - We have made the code available online and included a reference to the code in the paper.
2. The main body of the paper is written clearly and succinctly. I admire the authors' commitment to maximizing the mechanistic realism of the vector movement process through their hierarchical approach, and while I do think the data requirements are quite high in order to replicate the case study, this is likely not the only case for which it is applicable, though I would have appreciated a discussion of how the model could be modified for cases where such a thorough survey protocol was not possible (e.g. how can mail-out survey data be incorporated?).
 - Thank you for this comment. A major motivation for applying the hierarchical model is the difficulty to use "classical" data sources in many systems of interest. We have extended our explanation of this issue in both the introduction and discussion. Nonetheless, we have added a note on how classical survey data could be used if available (discussion section). Though a more detailed explanation might be desirable, this would have added more technicalities to the paper, which is rather long already. Since the use of classical data sources has been discussed extensively in the literature and classical stochastic gravity models can easily be integrated in our hierarchical framework, we have decided to keep the discussion on the application of mail-out data short.
3. The paper's appeal could be much more broad if more of the case study results were included in the main text, and the inclusion of validation results in the main text would provide a more honest picture of the demonstrated strength of the methodology. While I think the novelty of this method is sufficient for this work to be highly important in spite of only moderate demonstrable predictive power, I think that the results should be presented more transparently in text. The inclusion of only fitted data in in-text figures masks the predictive ability of the model, which is only visualized through predicted vs. observed plots in the supplement. In addition, the authors refrain from referring to any quantitative metrics of fit. I understand that the model does not conform to the assumptions of traditional R^2 metrics, but given that the alternative is visual inspection, I would argue that readers will interpret plots by visually estimating R^2 regardless of whether it is appropriate (e.g. perhaps the author's could provide a Pseudo- R^2 metric for a negative binomial GLM like Nagengke). Further, while I understand the full scope of validation presented in the supplement cannot be applied to other gravity models, I do think that some comparison of predictive ability to previous work should be made (how strong are predictions of source/destination pairs?) in order to assess the performance of this approach relative to other methods.
 - Thank you for these helpful suggestions! We have made considerable effort to address the comments:
 - We have moved several model and validation results from the Appendix to the main text (sections results and discussion).
 - We have taken efforts to provide R^2 values where possible (sections application, results, and discussion).
 - We have compared our R^2 values to values obtained in earlier comparable studies (section discussion).

- Due to the length of the paper, we could not move all desired validation results to the main text. Nonetheless, we have included more references to the additional validation results (discussion) to increase their visibility.
 - Though computing R^2 values for our predicted versus observed plots is easy (and, in fact, we did it) these values would be misleading as they *overestimate* the predictive quality of our model. We therefore decided to present only those metrics that are suited to accurately assess our model's predictive quality.
 - Please note that the common interpretation of R^2 as “explained variance” comes from the context of deterministic modelling, where the existence of a model matching error-free data perfectly is assumed. In this context, R^2 measures how close the presented model comes to this ideal. This contrasts with stochastic models, where the considered system is assumed to be inherently stochastic. Rather than getting close to a “correct” *value*, stochastic models seek to determine the correct *distribution* of the quantities of interest. Consider for example a model for rolling a die. A deterministic model may be used to estimate the expected value of the diced numbers (here 3.5). The accuracy of this prediction could be assessed via R^2 . A stochastic model, however, would provide the full distribution of the rolled numbers (here a discrete uniform distribution). The stochastic model yields much more insights, as it allows not only inference about the expected value but also, for example, about the median and other quantiles and the variance. The quality of such a model cannot be easily measured via R^2 . For this reason, we believe that R^2 yields only limited insights into our model's quality and that a more careful analysis as presented in this study is necessary. Therefore, we refrained from including R^2 for quantities where R^2 is not applicable.
4. Finally, I cannot fully evaluate the detailed mathematical proofs within the supplement, and would recommend the editor seek out a reviewer with the appropriate training in order to fully assess their validity.

Specific comments (in all cases I use the innermost line numbers):

Abstract:

5. I recommend including some quantitative metric of model fit for the case study here (like a Pseudo R^2).
- We have computed an R^2 value for variance-corrected mean boater counts at survey locations and also a pseudo R^2 for the entire model.

Introduction:

6. lines 88-91: isn't this easily circumvented by only inquiring about historical trips?
- This sentence was misleading. We have corrected it and included an example to make our argument clearer.

Model:

7. Overall, I find the schematic figures very helpful and the approach clear to follow.
8. Eqn 1 - c should be defined in text
- We have adjusted the text accordingly.
9. Lines 179-181: What about factors that have both a main effect and interaction?
- Thank you for pointing this out. This would be possible as well. We have adjusted the text.
10. Lines 204-207: I am not sure I follow this logic - why would the rarity of total agents of concern affect whether these agents travel independently?
- Thank you for making us aware of this issue. We have added further clarification and also included a note that rarity is not sufficient to justify independence.
11. Lines 217-219: Is there any evidence in the literature that most travellers make at the most one intermediate stop?

- Thank you for this question. The decision to focus on single-via paths in our model is not based on assumptions on how many stops an agent makes. In fact, even an agent taking the direct route may take a large number of breaks. Our focus on single-via paths is mainly due to technical limitations, because the number of locally optimal paths is typically too large to be handled. We have added this clarification to the text.

Model Fit:

12. I believe that some mention of the software used for fitting the model is necessary in the main text.
 - We have included references to the used software.

Application:

13. Lines 323-327: Was there a specific boat size cutoff used?
 - There was not a specific boat size cutoff used. Instead, we classified boats based on their type, which typically coincides with a specific size range. The main boat types we considered are provided in the referred section.
14. Lines 358-362: How correlated were the predictor variables tested across submodels? How well does the model distinguish highly correlated predictors? A correlation matrix or even some discussion on the relative identifiability as it relates to correlation strength would be useful in an appendix.
 - Thank you for these questions. The typical motivation for conducting a correlation analysis for predictor variables is to ensure that parameters are estimable. Since non-estimable parameters have large confidence intervals (see Yoo et al., 2014; reference included in the revised manuscript), the confidence intervals we computed suffice to assess whether estimability issues exist. Note that if enough data are available and/or the effects of predictors are large, parameters can be estimable despite correlations among predictors. Therefore, a thorough analysis of confidence intervals typically yields more insights than a correlation analysis. We have added this argument to the text (section 4.1.2) and included further discussion of parameter estimability in the manuscript (section 5.2).
15. Figure 4: Without the raw data, this figure does not seem very informative.
 - We have added traffic density estimates based on step function distributions without pre-defined shape to the Figure. These provide an image of the raw data.
 - Though we agree with the reviewer that it were beneficial if raw data could be visualized in this figure, we are unsure which representation of raw data the reviewer had in mind. A non-parametric estimate of the temporal traffic density (e.g. a histogram) is not applicable here, because surveys were conducted at different daytimes at different locations. For example, at locations with little traffic, surveys concentrated on the peak times, whereas at locations with higher traffic, survey shifts were extended to cover the early morning and the late night. Therefore, a simple histogram approach would overestimate the traffic at marginal times.
16. Lines 376-382: I think the inclusion of the parameter estimates in Appendix F, or a reduced version of it, is worth including in the main text.
 - We have shifted the Appendix to the main text.
17. Line 386: While this is more relevant to a subsequent model of invasion risk, I feel some discussion is necessary about how donor regions far from recipients will have higher propagule mortality and thus lower risk.
 - We have added a corresponding note to the discussion (future directions).
18. Figure 6 - It is unclear which circles correspond with A,B,C as there are multiple access points - perhaps colour code A B and C lakes to ease interpretation. Further, is there anyway to visualize uncertainty in these plots? perhaps using transparent inner and outer circles?
 - Thank you for these suggestions!
 - We have colour coded the circles corresponding to A, B, and C to show which sections belong together.

- We have added lighter circles depicting the 95th percentile of boater counts to provide an impression of the estimated uncertainty. Note that a low quantile would be 0 for all of the lakes, which is why we did not include “inner circles”.

19. Figure 7 - Similarly, a depiction of uncertainty would be useful here. Perhaps even as CI values on the map itself

- Thank you for this suggestion. Unfortunately, including a measure for uncertainty is difficult in this Figure, because the estimates were depicted via graduated colours. However, we have included a thorough analysis of the uncertainty associated with estimating traffic counts at roads in the model validation section (see Figure 8 in the revised manuscript).

Discussion:

20. Line 411- perhaps I'm unfamiliar with this terminology, but I would suggest 'recipient regions' to better match your wording of 'donor regions'

- We have made the terminology consistent.

21. Lines 415-417 - at times I find that the paper reads more like a demonstration of the utility of surveying along roads rather than the model itself. I think this could be improved by an additional discussion of how other sources of data that may already exist could also be included (e.g. mail-out surveys or surveys at lakes), and if partial information e.g. from booking systems can be somehow incorporated in cases where such a thorough survey is not possible.

- Thank you for this suggestion. We have added a corresponding note to the discussion.
- It is a major focus of our study to show how traffic estimates can be obtained even if booking records or mail-out survey data *cannot* be obtained. Consequently, the case that such data *are* available in sufficient quality is beyond the scope of our study. If data from booking records *were* available, these could be used to fit the gravity model directly. There are many studies showing how to do this. Incorporating the fitted gravity model in the hierarchical model would be straightforward.

22. Line 495 - remove 'they'

- We have removed the word.

23. Line 497 - again, this is more of an advantage of intensive surveying rather than the methodology presented, per se.

- Thank you for this comment. In the referred section, we argued that the limitation introduced by using a uniform compliance rate did not compromise our results. We did not intend to present this as an advantage of our method.

24. Line 500 - Is there a chance that the implementation of required stopping points along roads influenced traffic patterns?

- Thank you for this question. Compliant boaters were not fined at inspection stations regardless of whether their watercraft were infested. Furthermore, due to the structure of the British Columbian road network, the length of an inspection is typically short compared to the time needed to bypass an inspection station on an alternative route. For these reasons, it would rarely be rational for a boater to change their routing decision based on the locations of watercraft inspection stations. Consequently, we think that it is unlikely that the inspection stations influenced the traffic patterns significantly. Nonetheless, agents actively avoiding inspection stations could be an issue in other systems. We included a corresponding paragraph in the discussion.

25. Line 510 - Is there a way to estimate what fraction of included potential routes are unrealistic?

- Thank you for this question. Line 510 refers to the noise model accounting for boaters whose paths we did *not* include. In our model, this were 5% of the boaters. We clarified this in the text.

Supplement:

26. Line 26: Is there any potential to create gradations of route admissibility in order to limit the number of clearly impossible routes?
- Thank you for this question and the interesting idea. There are two motivations for considering route admissibility: (1) to model agents' inability to consider all potential routes ("bounded rationality") and (2) to make quantitative assessment of route quality computationally feasible. Note that the number of possible routes is extremely large already in small road networks, making it impossible to process all paths in reasonable time even with the fastest computers.
 - It would be possible to introduce gradations of route admissibility to improve the model for bounded rationality. For example, some agents may be able to consider more alternatives than others. This could be accounted for by considering gradations of route admissibility.
 - When it comes to deciding which routes are computed and which are not, we cannot make a partial decision. Either we compute and consider a route or not. Therefore, gradations of route admissibility cannot help making the model more computationally efficient.
 - Improving the route choice model by introducing gradated route admissibility would be an interesting task for future research but is beyond the scope of this paper. Nonetheless, it is unlikely that we will ever be able to cover all routes that agents may take. Therefore, there will always be a need for a null model for agents travelling on unknown routes. It may be possible to develop a more realistic null model than the model we used in this study. However, this is beyond the scope of our paper.
27. Line 33: How important are multiple trips by the same agent? Does a large fraction of multiple trips affect the assumptions of independence?
- Thank you for this excellent question. Multiple trips by the same agent are infrequent in large-scale systems. Nonetheless, it is possible that the same agent makes multiple trips, which is a potential source of correlations in data. However, it is difficult to know the sign and strength of these correlations (does a boater prefer to travel to the same lake visited before or do they want to try a new one?). If corresponding data were available, this would be a worthwhile subject for future research. In the absence of a model for the correlations, we may continue to work under the independence assumption. Nonetheless, this is a potential model limitation.
28. Line 79: How might this varying coarseness impact the management implications of this approach? e.g. I would think it becomes less clear how to target non-BC sites for surveillance/control measures.
- Thank you for pointing this out. We were interested in results that would facilitate invasive species management in BC. We have added clarification that the road network would need to be adjusted if inspections at locations outside of BC were of interest as well.
29. Lines 81-86: I think this information is worth moving to main text.
- We have moved this information to the main text.
30. Line 99: affect—> effect
- We have corrected this error.
31. Line 103: Again, prefer 'recipient regions'
- We have made the notation consistent within this paragraph.
32. Line 104: Perhaps I'm misunderstanding, but I think the word 'not' is superfluous here
- This is correct. We have removed the word.
33. Line 116-117: As mentioned above, I think facilitating implementation is of key importance in keeping with the principles of Open Science, and that details and code should be made available.
- We have added a reference to the code.
34. Line 236: I would change the notation d to a different letter so that it is not confused with distance in the gravity model equations in the main text.
- We have replaced the letter with a capital D .
35. Eqn A5: What is the meaning of the tilde in these equations?

- We have used the tilde to distinguish distinct variables with related meaning. The tilde in itself has no meaning but helps to prevent potential confusion if too many different symbols would have been used.
36. Lines 464-465: I think this information should be included in the methods of the main text.
- We have moved this information to the main text.
37. Line 491: loose—>lose
- We have corrected this error.
38. Line 494: I would recommend predictive performance on a validation set as a better alternative to AIC (e.g. choosing the model that minimizes RMSE).
- Thank you for this interesting idea. Selecting the model based on its predictive quality would indeed be a potential alternative approach and would circumvent issues arising from independence approximations. Nonetheless, selecting the model based on RMSE has some drawbacks as well.
 - Minimizing RMSE is motivated from a phenomenological modelling perspective: the model is optimized to predict the data best, thereby disregarding the model's mechanistic validity. In contrast, a maximum likelihood-based approach such as AIC takes the modelled mechanisms into account. The maximum likelihood approach is therefore better suited to fit mechanistic models such as ours.
 - Minimizing RMSE would require us to focus on a specific quantity of interest and choose, for example, the model that best estimates the mean traffic at inspection locations. The selected model, however, might be poor at predicting other quantities of interest, such as the number of boaters arriving at lakes. This applies also to other statistical quantities of interest, such as the variance, the median, or the n^{th} quantile of the boater flow. As the likelihood is a better measure for mechanistic validity than RMSE, a model selected based on AIC may be more versatile.
 - Though applying RMSE to a validation data set provides a certain protection against overfitting, model selection based on RMSE has the potential to include unnecessary parameters. Due to the asymptotic properties of the maximum likelihood estimator, the likelihood values on the fitting and the validation set will coincide if the data set is sufficiently large. If RMSE is computed based on a likelihood-based pseudo- R^2 , such as the metric by Nagelkerke, then RMSE will always select for the model with maximal likelihood without punishing excess parameters. As a consequence, unimportant covariates might be included in the model, and parameters might not be estimable.
 - Due to the reasons provided above, we decided to continue using AIC for model selection.
39. Line 508: I would be careful with this justification, as the mechanistic separation of traffic flow and route choice is one of the arguments for building the model in the introduction, and no clear advantage has been shown for this model's ability to predict the correct propagule pressure relative to less mechanistic alternatives.
- We agree. We have removed the sentence.
40. Lines 522-528: I think this is worth moving to the main text.
- We have moved the section to the main text.
41. Line 535: The separability of predictive factors is still useful in order to extrapolate the model predictions across time and space. How sensitive would the model be to changes in correlation structure of the modelled predictors?
- This is a very interesting comment. The analysis of parameter confidence intervals suggests that correlations would not hinder extrapolation of model predictions. Nonetheless, changes in the correlation structure could have an effect, which would be interesting to investigate in future studies.
42. Table A3: I would recommend including this table in the main text, and would recommend centring and scaling the predictor variables so that their relative magnitudes are meaningful.

- We have moved the table to the main text.
 - We chose the units of the predictor variables so that they were of the same order of magnitude. Further rescaling might ease the comparison of parameters but would also make the predictor variables dimensionless. We believe that it is easier to interpret the parameters if they refer to predictors with dimensions. Therefore, we did not apply further rescaling.
43. Line 564: I would like to see the raw data plotted on Figure 4.
- Please refer to our response to point 15.
44. Line 962: This is where I think RMSE and Pseudo-R² metrics would be advantageous to present. Regardless of their validity, the reader is going to compare the points to a 1-to-1 line in a way that approximates R² anyway. It also facilitates the interpretation of the relative amount of error in each component of the model, and RMSE could be used for model comparison as an alternative to AIC given that AIC assumptions are also not met.
- Thank you for this suggestion. In point 3 we explain why we are hesitant to provide R² metrics where they are not valid, and under point 38 we list the drawbacks of RMSE that let us decide to continue working with AIC for model selection.
 - It is an interesting idea to use R² as to assess the relative amount of error in each model component. However, we think that the R² values would not permit quantitative interpretation in our case. For qualitative insights, graphical comparison may suffice to determine the quantities that are most challenging to estimate.
 - Though we agree that R² often leads to similar conclusions as graphical comparison, it is not clear that the results coincide completely. For example, getting an impression for the *squared* error from a graph may require considerable training. Furthermore, it is possible and intuitive to consider the *relative* error visually, which may differ significantly from R². The relative error is a better measure for model performance in our case. Note in addition that in contrast to visual representations, R² values have a quantitative interpretation that would be misleading in our case.
45. Lines 1007-1008: This result is particularly troubling, as the source and destination of vectors is key for predicting the spread of invasions. This is where a comparison between this model's predictive power and that of previous gravity models would strengthen the paper.
- We have included a comparison with earlier gravity models in the discussion section.
 - For predicting the spread of invasions, the number of infested vectors arriving at destination lakes is key. Therefore, earlier models fitted to historical invasion data were (implicitly) optimized to accurately predict the vector inflow to lakes. Since these models were often evaluated with respect to their ability to mimic the invasion process rather than to model traffic accurately, a comparison of such models with ours is difficult.
 - Even if our model does not yield more accurate estimates than earlier approaches, it is applicable in large-scale systems where the data needed to fit "classical" gravity models are not available. Therefore, we see the main advantage of our model in its broad applicability rather than its potentially higher accuracy.
46. Figure A3: For transparency, I think this needs to be in the main text. Do the author's have any ideas on the causes of major outliers?
- We have included more validation results in the main text and also added further references to this appendix. However, due to the length of the paper, we could not include this figure and its interpretation in the main text.
 - Some of the outliers may be due to the spatial resolution of the data and difficulties to determine the exact access points of boaters. In addition, some lakes have a long touristic history that could not be fully captured by the covariates. However, often a very thorough analysis of individual observations would be necessary to find the causes of outliers. This would be a worthwhile task for future research and could improve the model's accuracy further.
47. Lines 1080-1086: I think this paragraph would also fit well in the discussion after moving more of the case study methods and results to the main text.

- We have added an abridged version of this paragraph to the discussion and also included another reference to this appendix.

48. Line 1090: issued—>issues

- We have corrected this error.

49. Line 1092: But would this make the model's predictive power sensitive to fluctuations in population size over time? Can this model effectively forecast vector movement with changing human population projections?

- Thank you for pointing this out! The model in the revised manuscript has one parameter less. Nonetheless, the estimability issue persists, and we have given it more thorough consideration in the discussion.
- The model should be applied carefully when it is transferred to systems with many lakes in highly populated regions. We have added this caveat to the discussion.
- Both the habits of boaters and the covariates used in the model are subject to change over time. Therefore, the model may not be suitable to make long-time predictions. However, this is a caveat to most traffic models, and changes in covariates occur typically on a slower scale than the invasion process. If necessary, we would suggest refitting the model after a moderate time to account for changes.

Reviewer: 2

The study follows two goals, i) to present a hybrid gravity and route choice model for estimating traffic in a transportation network, and ii) to apply this model for investigating the potential invasion of Zebra and Quagga mussels into British Columbia. This is an interesting and stimulating study. The proposed hybrid model is an interesting contribution to traffic modelling. The model is also well presented and the supplementary information contains my model details. However, I found some issues, mostly in the application of the model which is only weakly analysed (see comments below). I would recommend publication of this study when my these issues can be resolved by the authors.

Comments:

Section 2.2 Route change model.

- I had some problems following the rationale behind the model choices here, possibly because I am not familiar with Fischer (2019).
 - We have made efforts to make this section more understandable. However, an in-depth elaboration of traffic modelling options is beyond the scope of our paper. We would like to point the interested reader to the cited references.
- 213-214 "We illustrate this concept.. in Fig.2". The figure does not mention the parameter delta, so how can it illustrate this concept?
 - We have adjusted the figure caption to describe the role of the parameter delta.
- 216: What is the difference between this rule (using parameter gamma) and the previous rule (using parameter delta). At first glance they seem very similar. Maybe an improved Fig.2 can help here.
 - We have added further clarification to the text.
- 217: "we focus single-via paths": Please state clearly, what you mean by "focus" in this context. Do you really exclude all paths that contain more than one intermediate step-over location? If yes, can you please give more arguments for this approximation.
 - We have adjusted the text to clarify the ambiguous phrase and added further arguments.

- Eq (2): by this equation longer routes automatically are down-regulated. Do we then still need the other extra rules (including delta, gamma, and eta)?
 - We have modified the text contrasting the two rules to answer this question more clearly.

Section 2.3 Temporal patterns

- While I can see the elegance to include the temporal dynamics in the general hierarchical framework (Fig. 1), I have a problem to comprehend the relevance of diurnal traffic patterns for this specific application. Do the authors really believe that a resolution to a day-night temporal scale is important to understand the invasion patterns into BC? If yes, can you please give some more arguments? Note that this is different on whether or not a von-Mises function gives the best description of the temporal pattern (as discussed in detail).
 - Thank you for this question. We have added further clarification to the text.
 - The reason why it was necessary to consider daily traffic patterns is that surveys were conducted at different daytimes at different locations. Thereby, surveys concentrated on the peak times at locations with little traffic, whereas survey shifts were extended to cover the early morning and the late night at locations with higher traffic. Since there is a strong dependence of traffic on the daytime (see the section results), ignoring temporal traffic variations would have introduced a bias underestimating the traffic at busy locations. Furthermore, we were not able to conduct many surveys around the clock. Therefore, we needed a way to estimate which portion of the traffic we missed by not surveying at night.
 - Besides being necessary to fit the model, the daily distribution of traffic is of interest to invasive species managers who seek to inspect as many high-risk watercraft for invasive species as possible. Knowing the temporal distribution of traffic helps managers to determine the optimal timing for inspections.
- A simple test to demonstrate the prediction-gain by inclusion of the temporal aspect would be to compare model runs with, and without, the temporal components. Would we see differences?
 - This is an interesting suggestion (and could be done for the other submodels as well). However, we would like to bring the following points to the reviewer's attention:
 - The hybrid model could not be fitted without an (at least implicit) temporal component, because the survey times differed between survey shifts.
 - The error in one model component (e.g. the travel time model) would be partially compensated by a different model component (e.g. the route choice model). Though this effect increases the accuracy of model predictions, it can lead to misleading parameter estimates and interpretations and decreases the model's portability to other systems.
 - Due to the reasons outlined above, we think that it is important to consider the temporal component even if predictions based on a null model would not differ a lot from the predictions of the full model. Therefore, we have not included this analysis.
- Fig.7, legend "road's lanes are coloured separately to depict traffic in different driving directions": Can you explain this better? I do not understand this sentence and cannot see different directions in the figure.
 - Thank you for pointing us towards this issue. We have edited the figure caption and added road outlines in the figure to show the two lanes more clearly.

Results:

- After having been introduced to the (rather involved) model, I was somewhat disappointed that the only shown results are three or four geographic pictures. I am missing a discussion of some broader patterns. Some questions along these lines are: Do we have general distance dependencies in the model outcome? Can we aggregate model outcomes over some variables to learn something about the invasion process?
 - This are excellent questions! We have included more model results and interpretation in the main text.
 - Please note that though the presented model can serve as a major component of an invasion model, the hybrid model does not cover all stages of the invasion. This often makes it difficult to answer broader questions satisfactory. For example, though gravity models include a general

distance dependency by construction, the transport survival of propagules depends on the transport distance as well. Therefore, the gravity model does not suffice to measure the distance dependency in full. We have highlighted the need to combine our model with further submodels in the discussion.

- Furthermore I miss any kind of sensitivity analysis. The Supplementary information contains a wealth of information on the formal aspects of how the fitting was performed, but I miss more simple information on how the model results depend on the many model parameters and model choices?
 - Thank you for this suggestion. We have added further results showing the impact of the covariates on the results to the main text. We agree that sensitivity analysis is usually a crucial part of model analysis. However, we believe that this is different in our case. We would like to explain our reasoning below.
 - There are two major motivations for conducting a sensitivity analysis: (1) to assess the potential impact of parameter uncertainty on results and (2) to assess the impact of potential changes in modelled mechanisms on results. In other words, sensitivity analysis is important to determine how robust a result is to error and future changes of the modelled system. However, we argue that sensitivity analysis is not well suited to achieve goal (1) in our case and that goal (2), though a worthwhile task for future research, would require an analysis that is beyond the scope of this paper. Below we provide details.
- (1) Sensitivity analysis is well suited to assess the implications of parameter uncertainty if parameters are estimated individually from separate data sets. A high sensitivity of predictions is a warning sign that results could be subject to significant error if parameters are not estimated precisely. However, when model parameters are fitted simultaneously to the same type of data that the model seeks to predict, things are different. In this case, impactful parameters will have small confidence intervals (i.e. be estimated precisely), because small changes in the parameters would change predictions strongly and thus worsen the model fit significantly. That is, if the model is sensitive to a parameter, this parameter will be estimated very precisely, which in turn will compensate potential issues resulting from sensitivity. Since our model is fitted to the same sort of data the model seeks to predict, a sensitivity analysis would provide very limited insights on uncertainty.
- (2) When the goal is to assess the implications of changes in mechanisms, two challenges arise: (A) a specific result of interest must be selected for the analysis and (B) the model parameters must be understood mechanistically. These challenges combined make sensitivity analysis difficult in our case, as we will show now.
 - (A) Changing a parameter may increase one prediction and decrease another. For example, a stronger preference for short routes may increase the traffic at one location and decrease the traffic at another location. A thorough sensitivity analysis would therefore need to consider all results of interest separately. Though this could lead to insightful results and would be an interesting subject for future research, this is beyond the scope of our paper. Alternatively, we could try to identify summarising quantities that represent all results. In our case, this could be the total number of high-risk boaters driving to BC each day. Aggregation, however, limits some insights. The total number of travelling boaters, for example, is independent of boaters' compliance with surveys, their travel timing, and their route choice. Consequently, only the gravity model would be analysed with this approach. Analysing the sensitivity of the gravity model, in turn, is subject to challenge (B) outlined below.
 - (B) If the goal is to study implications of changing mechanisms, it is necessary that these mechanisms are included in the model explicitly. Our gravity model, however, is a phenomenological model. For example, while the gravity model accounts for higher attractiveness of lakes with touristic facilities, it does not model boaters' destination choice explicitly. As a consequence, a change in mechanism is difficult to model by changing one parameter only. For example, a trend of increased attractiveness of lakes with touristic facilities would likely coincide with a trend of decreased attractiveness of lakes without these facilities. Since our model does not include this trade-off directly, multiple parameters would need to be changed simultaneously to consider the scenario. This, however, would exceed the scope of a classical sensitivity analysis.
- Due to the reasons outlined above, we believe that a classical sensitivity analysis would either yield limited insights or require extensive computation and discussion. A thorough scenario

analysis would be very interesting, and we hope that this could be done in a future paper, but this involved analysis is beyond the scope of our paper.

- We have included an abridged version of the above argument in the paper (section discussion/model validation and accuracy).

Appendix B

Response Letter

Below please find our answers to the reviewer comments. For easier reference, we have numbered the comments. Our answers are indented and written in blue font below the comments.

Reviewer: 1

I was pleased with the authors' thorough consideration of all of my comments, and believe that the changes that they have made have greatly strengthened their manuscript. I note that I have only examined their responses to my own comments.

I am happy that the authors now provide the code to make their results more transparent and reproducible, that their validation results are more visible, and that they situate their findings within the existing gravity model literature. I also appreciate the increased space given to justifying their particular modelling approach versus conventional approaches.

I have a few specific responses to their changes below:

- 15. Figure 4: I see now that it is not as simple as plotting a histogram, thank you for the clarification.
- 38. RMSE discussion: I respect the authors' choice to perform model selection via AIC, but don't fully agree that this approach ensures the selected model is mechanistically appropriate (from what I've understood of their approach, it would only ensure that the error distributions from which likelihood is computed are appropriate). I also would expect that RMSE could be summed across quantities of interest (analogous to how the likelihood is computed jointly), thereby not requiring the authors to focus on a single quantity upon which to assess model fit. Further, I disagree that RMSE on a validation set is susceptible to overfitting. If some additional noise term is fit to, say, a multiple regression, there is no reason to expect the validation data to have the same spurious correlation with that noise term, so it should not improve RMSE estimates. In any case, I acknowledge that neither metric is perfect, and respect the choice of AIC.
 - Thank you for the valuable insight and for respecting our choice to use AIC for model selection.
- 44. I recommend the authors calculate a relative error metric such as relative absolute error (RAE) for the cases in which they believe it is more appropriate than squared error.
 - Thank you for this recommendation. We have computed the mean absolute errors relative to the predicted standard deviations for the quantities depicted in Figure A3 in the supplementary appendix. We report the values in the corresponding figure caption as well as in the results section of Supplementary Appendix F. Furthermore, we discuss the results in the discussion section of Supplementary Appendix F.
- Spelling and grammar:
 - line 258 “needs” not need
 - line 322: prefer “improve” to “get better at”
 - line 424: prefer “sharp” to “spiky”
 - Table 1,2: table column headers seem buggy on my version
 - lines 684-685: prefer donor and recipient “regions” to maintain consistency
 - Thank you for pointing out these issues. We have corrected the errors.